# Functional repertoire convergence of distantly related eukaryotic plankton lineages abundant in the sunlit ocean

## Graphical abstract

## Authors

Tom O. Delmont, Morgan Gaia, Damien D. Hinsinger, ..., Eric Pelletier, Patrick Wincker, Olivier Jaillon

## Correspondence

tom.delmont@genoscope.fr

## In brief

Delmont et al. use nearly 300 billion metagenomic reads to characterize the genomic content of some of the most abundant and widespread eukaryotic populations in the sunlit ocean. This large genomic resource covers taxa underrepresented in our culture portfolio and exposes a functional convergence of distantly related eukaryotic plankton lineages.

## Highlights

- Nearly 300 billion metagenomic reads were co-assembled from plankton in the oceans

- Hundreds of eukaryotic environmental genomes were characterized and curated

- These genomes better represent eukaryotic plankton compared to cultivation

- These genomes reveal a functional convergence of distantly related eukaryotes

 Delmont et al., 2022, Cell Genomics 2, 100123
May 11, 2022 © 2022 The Author(s).

# Cell Genomics

CellPress

## Article

# Functional repertoire convergence of distantly related eukaryotic plankton lineages abundant in the sunlit ocean

Tom O. Delmont,[1,2,9,*] Morgan Gaia,[1,2] Damien D. Hinsinger,[1,2] Paul Frémont,[1,2] Chiara Vanni,[3] Antonio Fernandez-Guerra,[4] A. Murat Eren,[5] Artem Kourlaiev,[1,2] Leo d'Agata,[1,2] Quentin Clayssen,[1,2] Emilie Villar,[1] Karine Labadie,[1,2] Corinne Cruaud,[1,2] Julie Poulain,[1,2] Corinne Da Silva,[1,2] Marc Wessner,[1,2] Benjamin Noel,[1,2] Jean-Marc Aury,[1,2] Tara Oceans Coordinators, Colomban de Vargas,[2,6] Chris Bowler,[2,7] Eric Karsenti,[2,6,8] Eric Pelletier,[1,2] Patrick Wincker,[1,2] and Olivier Jaillon[1,2]

[1]Génomique Métabolique, Genoscope, Institut François-Jacob, CEA, CNRS, Université d'Evry, Université Paris-Saclay, 91057 Evry, France
[2]Research Federation for the Study of Global Ocean Systems Ecology and Evolution, FR2022/Tara GOSEE, 75016 Paris, France
[3]Microbial Genomics and Bioinformatics Research Group, Max Planck Institute for Marine Microbiology, Bremen, Germany
[4]Lundbeck Foundation GeoGenetics Centre, GLOBE Institute, University of Copenhagen, Copenhagen, Denmark
[5]Helmholtz Institute for Functional Marine Biodiversity at Oldenburg, Germany
[6]Sorbonne Université and CNRS, UMR 7144 (AD2M), ECOMAP, Station Biologique de Roscoff, Roscoff, France
[7]Institut de Biologie de l'ENS, Département de Biologie, École Normale Supérieure, CNRS, INSERM, Université PSL, Paris, France
[8]Directors' Research, European Molecular Biology Laboratory, Heidelberg, Germany
[9]Lead contact
*Correspondence: tom.delmont@genoscope.fr

## SUMMARY

Marine planktonic eukaryotes play critical roles in global biogeochemical cycles and climate. However, their poor representation in culture collections limits our understanding of the evolutionary history and genomic underpinnings of planktonic ecosystems. Here, we used 280 billion *Tara* Oceans metagenomic reads from polar, temperate, and tropical sunlit oceans to reconstruct and manually curate more than 700 abundant and widespread eukaryotic environmental genomes ranging from 10 Mbp to 1.3 Gbp. This genomic resource covers a wide range of poorly characterized eukaryotic lineages that complement long-standing contributions from culture collections while better representing plankton in the upper layer of the oceans. We performed the first, to our knowledge, comprehensive genome-wide functional classification of abundant unicellular eukaryotic plankton, revealing four major groups connecting distantly related lineages. Neither trophic modes of plankton nor its vertical evolutionary history could completely explain the functional repertoire convergence of major eukaryotic lineages that coexisted within oceanic currents for millions of years.

## INTRODUCTION

Plankton in the sunlit ocean contribute about half of Earth's primary productivity, impacting global biogeochemical cycles and food webs.[1,2] Plankton biomass appears to be dominated by unicellular eukaryotes and small animals[3–6] including a phenomenal evolutionary and morphological biodiversity.[5,7,8] The composition of planktonic communities is highly dynamical and shaped by biotic and abiotic variables, some of which are changing abnormally fast in the Anthropocene.[9–11] Our understanding of marine eukaryotes has progressed substantially in recent years with the transcriptomic (e.g.,[12,13]) and genomic (e.g.,[14–16]) analyses of organisms isolated in culture and the emergence of efficient culture-independent surveys (e.g.,[17,18]). However, most eukaryotic lineages' genomic content remains uncharacterized,[19,20] limiting our understanding of their evolution, functioning, ecological interactions, and resilience to ongoing environmental changes.

Over the last decade, the *Tara* Oceans program has generated a homogeneous resource of marine plankton metagenomes and metatranscriptomes from the sunlit zone of all major oceans and two seas.[21] Critically, most of the sequenced plankton size fractions correspond to eukaryotic organismal sizes, providing a prime dataset to survey genomic traits and expression patterns from this domain of life. More than 100 million eukaryotic gene clusters have been characterized by the metatranscriptomes, half of which have no similarity to known proteins.[5] Most of them could not be linked to a genomic context,[22] limiting their usefulness to gene-centric insights. The eukaryotic metagenomic dataset (the equivalent of ~10,000 human genomes) on the other hand has been partially used for plankton biogeographies,[23,24] but it remains unexploited for the characterization of genes and genomes due to a lack of robust methodologies to make sense of its diversity.

Genome-resolved metagenomics[25] has been extensively applied to the smallest *Tara* Oceans plankton size fractions,

unveiling the ecology and evolution of thousands of viral, bacterial, and archaeal populations abundant in the sunlit ocean.[26–31] This approach may thus be appropriate also to characterize the genomes of the most abundant eukaryotic plankton. However, very few eukaryotic genomes have been resolved from metagenomes thus far,[26,32–35] in part due to their complexity (e.g., high density of repeats[36]) and extended size[37] that might have convinced many of the unfeasibility of such a methodology. With the notable exception of some photosynthetic eukaryotes,[26,32,35] metagenomics is lagging far behind cultivation for eukaryote genomics, contrasting with the two other domains of life. Here we fill this critical gap using hundreds of billions of metagenomic reads generated from the eukaryotic plankton size fractions of *Tara* Oceans and demonstrate that genome-resolved metagenomics is well suited for marine eukaryotic genomes of substantial complexity and length exceeding the emblematic gigabase. We used this new genomic resource to place major eukaryotic planktonic lineages in the tree of life and explore their evolutionary history based on both phylogenetic signals from conserved gene markers and present-day genomic functional landscape.

## RESULTS AND DISCUSSION

### A new resource of environmental genomes for eukaryotic plankton from the sunlit ocean

We performed the first, to our knowledge, comprehensive genome-resolved metagenomic survey of microbial eukaryotes from polar, temperate, and tropical sunlit oceans using 798 metagenomes (265 of which were released through the present study) derived from the *Tara* Oceans expeditions. They correspond to the surface and deep chlorophyll maximum layer of 143 stations from the Pacific, Atlantic, Indian, Arctic, and Southern Oceans, as well as the Mediterranean and Red Seas, encompassing eight eukaryote-enriched plankton size fractions ranging from 0.8 μm to 2 mm (Figure 1; Table S1). We used the 280 billion reads as inputs for 11 metagenomic co-assemblies (6–38 billion reads per co-assembly) using geographically bounded samples (Figure 1; Table S2), as previously done for the *Tara* Oceans 0.2–3 μm size fraction enriched in bacterial cells.[26] We favored co-assemblies to gain in coverage and optimize the recovery of large marine eukaryotic genomes. However, it is likely that other assembly strategies (e.g., from single samples) will provide access to genomic data our complex metagenomic co-assemblies failed to resolve. In addition, we used 158 eukaryotic single cells sorted by flow cytometry from seven *Tara* Oceans stations (Table S2) as input to perform complementary genomic assemblies (STAR Methods).

We thus created a culture-independent, non-redundant (average nucleotide identity <98%) genomic database for eukaryotic plankton in the sunlit ocean consisting of 683 metagenome-assembled genomes (MAGs) and 30 single-cell genomes (SAGs), all containing more than 10 million nucleotides (Table S3). These 713 MAGs and SAGs were manually characterized and curated using a holistic framework within anvi'o[38,39] that relied heavily on differential coverage across metagenomes (STAR Methods and supplemental information). Nearly half the MAGs did not have vertical coverage >10× in any of the metagenomes, emphasizing the relevance of co-assemblies to gain

sufficient coverage for relatively large eukaryotic genomes. Moreover, one-third of the SAGs remained undetected by *Tara* Oceans' metagenomic reads, emphasizing cell sorting's power to target less abundant lineages. Absent from the MAGs and SAGs are DNA molecules physically associated with the focal eukaryotic populations, but that did not necessarily correlate with their nuclear genomes across metagenomes or had distinct sequence composition. They include chloroplasts, mitochondria, and viruses generally present in multi-copy. Finally, some highly conserved multi-copy genes such as the 18S rRNA gene were also missing due to technical issues associated with assembly and binning, following the fate of 16S rRNA genes in marine bacterial MAGs.[26]

This new genomic database for eukaryotic plankton has a total size of 25.2 Gbp and contains 10,207,450 genes according to a workflow combining metatranscriptomics, *ab initio,* and protein-similarity approaches (STAR Methods). Estimated completion of the *Tara* Oceans MAGs and SAGs averaged to ~40% (redundancy of 0.5%) and ranged from 0.0% (a 15-Mbp-long Opisthokonta MAG) to 93.7% (a 47.8-Mbp-long Ascomycetes MAG). Genomic lengths averaged to 35.4 Mbp (up to 1.32 Gbp for the first giga-scale eukaryotic MAG, affiliated to *Odontella weissflogii*), with a GC-content ranging from 18.7% to 72.4% (Table S3). MAGs and SAGs are affiliated to Alveolata (n = 44), Amoebozoa (n = 4), Archaeplastida (n = 64), Cryptista (n = 31), Haptista (n = 92), Opisthokonta (n = 299), Rhizaria (n = 2), and Stramenopiles (n = 174). Only three closely related MAGs could not be affiliated to any known eukaryotic supergroup (see the phylogenetic section). Among the 713 MAGs and SAGs, 271 contained multiple genes corresponding to chlorophyll *a-b* binding proteins and were considered phytoplankton (Table S3). Genome-wide comparisons with 484 reference transcriptomes from isolates of marine eukaryotes (the METdb database[40] that improved data from MMETSP[12] and added new transcriptomes from *Tara* Oceans; see Table S3) linked only 24 of the MAGs and SAGs (~3%) to a eukaryotic population already in culture (average nucleotide identity >98%). These include well-known Archaeplastida populations within the genera *Micromonas*, *Bathycoccus*, *Ostreococcus*, *Pycnococcus*, *Chloropicon*, and *Prasinoderma* and a few taxa among Stramenopiles (e.g., the diatom *Minutocellus polymorphus*) and Haptista (e.g., *Phaeocystis cordata*). Among this limited number of matches, MAGs represented a nearly identical subset of the corresponding culture genomes (Figure S1, Table S4). Overall, we found metagenomics, single-cell genomics, and culture highly complementary with very few overlaps for marine eukaryotic plankton's genomic characterization.

The MAGs and SAGs recruited 39.1 billion reads with >90% identity (average identity of 97.4%) from 939 metagenomes, representing 11.8% of the *Tara* Oceans metagenomic dataset dedicated to unicellular and multicellular organisms ranging from 0.2 μm to 2 mm (Table S5). In contrast, METdb with a total size of ~23 Gbp recruited fewer than 7 billion reads (average identity of 97%), indicating that the collection of *Tara* Oceans MAGs and SAGs reported herein better represents the diversity of open ocean eukaryotes compared to transcriptomic data from decades of culture efforts worldwide. The majority of *Tara* Oceans metagenomic reads were still not recruited, which could be

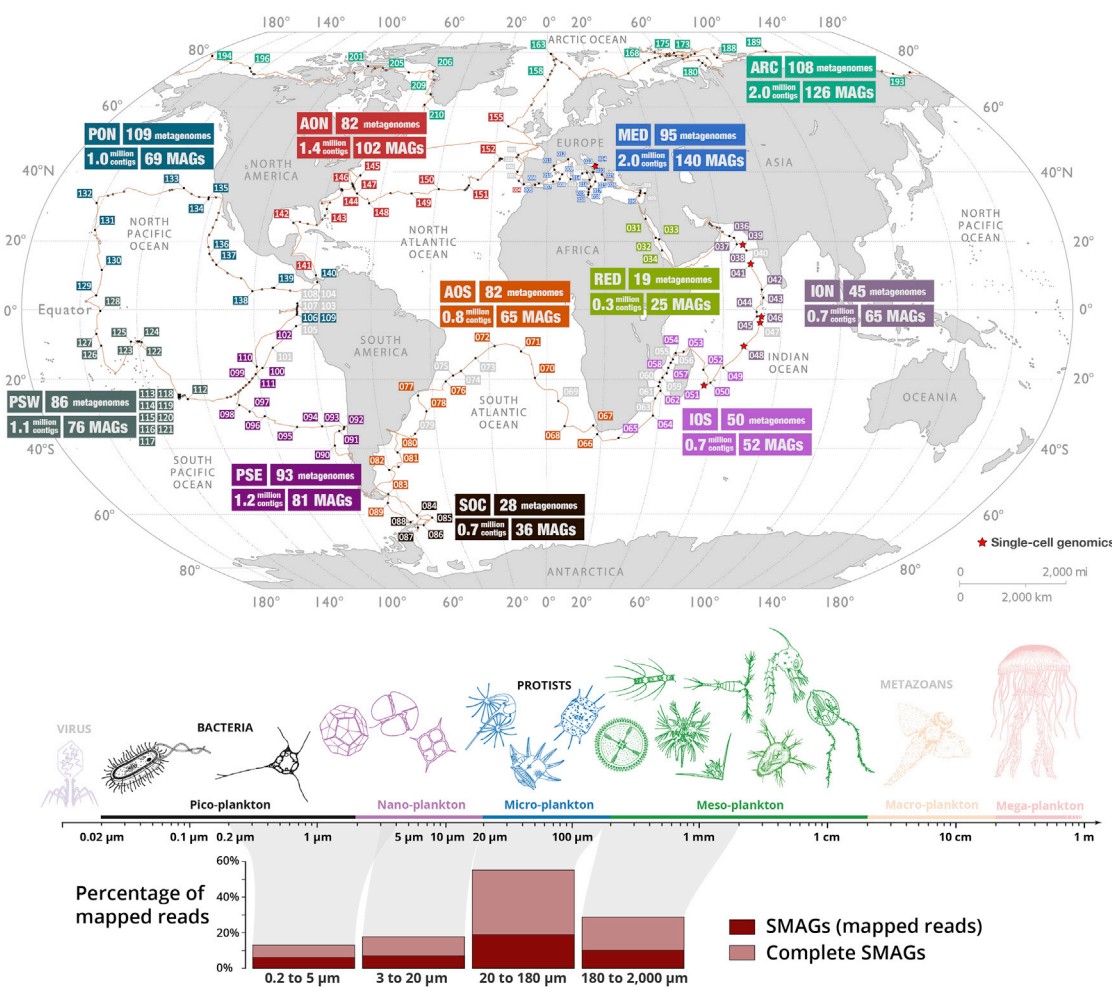

**Figure 1. A genome-resolved metagenomic survey dedicated to eukaryotes in the sunlit ocean**

The map displays *Tara* Oceans stations used to perform genome-resolved metagenomics, summarizes the number of metagenomes, contigs longer than 2,500 nucleotides, and eukaryotic MAGs characterized from each co-assembly, and outlines the stations used for single-cell genomics. ARC: Arctic Ocean; MED: Mediterranean Sea; RED: Red Sea, ION: Indian Ocean North; IOS: Indian Ocean South; SOC: Southern Ocean; AON: Atlantic Ocean North; AOS: Atlantic Ocean South; PON: Pacific Ocean North; PSE: Pacific South East; PSW: Pacific South West. The bottom panel summarizes mapping results from the MAGs and SAGs across 939 metagenomes organized into four size fractions. The mapping projection of complete MAGs and SAGs is described in the STAR Methods and supplemental information.

explained by eukaryotic genomes that our methods failed to reconstruct, the occurrence of abundant bacterial, archaeal, and viral populations in the large size fractions we considered,[41–43] and the incompleteness of the MAGs and SAGs. Indeed, with the assumption of correct completion estimates, complete MAGs and SAGs would have recruited ~26% of all metagenomic reads, including >50% of reads for the 20–180 μm size fraction alone due in part to an important contribution of hundreds of large copepod MAGs abundant within this cellular range (see Figure 1 and Table S5).

## Expanding the genomic representation of the eukaryotic tree of life

We then determined the phylogenetic distribution of the new ocean MAGs and SAGs in the tree of eukaryotic life. METdb was chosen as a taxonomically curated reference transcriptomic

database from culture collections, and the two largest subunits of the three DNA-dependent RNA polymerases (six multi-kilo-base genes found in all modern eukaryotes and hence already present in the last eukaryotic common ancestor) were selected. These genes are highly relevant markers for the phylogenetic inference of distantly related microbial organisms[44] and contributed to our understanding of eukaryogenesis.[45] They have long been overlooked to study the eukaryotic tree of life, possibly because automatic methods are currently missing to effectively identify each DNA-dependent RNA polymerase type prior to performing the phylogenetic analyses. Here, protein sequences were identified using hidden markov models (HMMs) dedicated to the two largest subunits for the MAGs and SAGs (n = 2,150) and METdb reference transcriptomes (n = 2,032). These proteins were manually curated and linked to the corresponding DNA-dependent RNA polymerase types for each subunit using

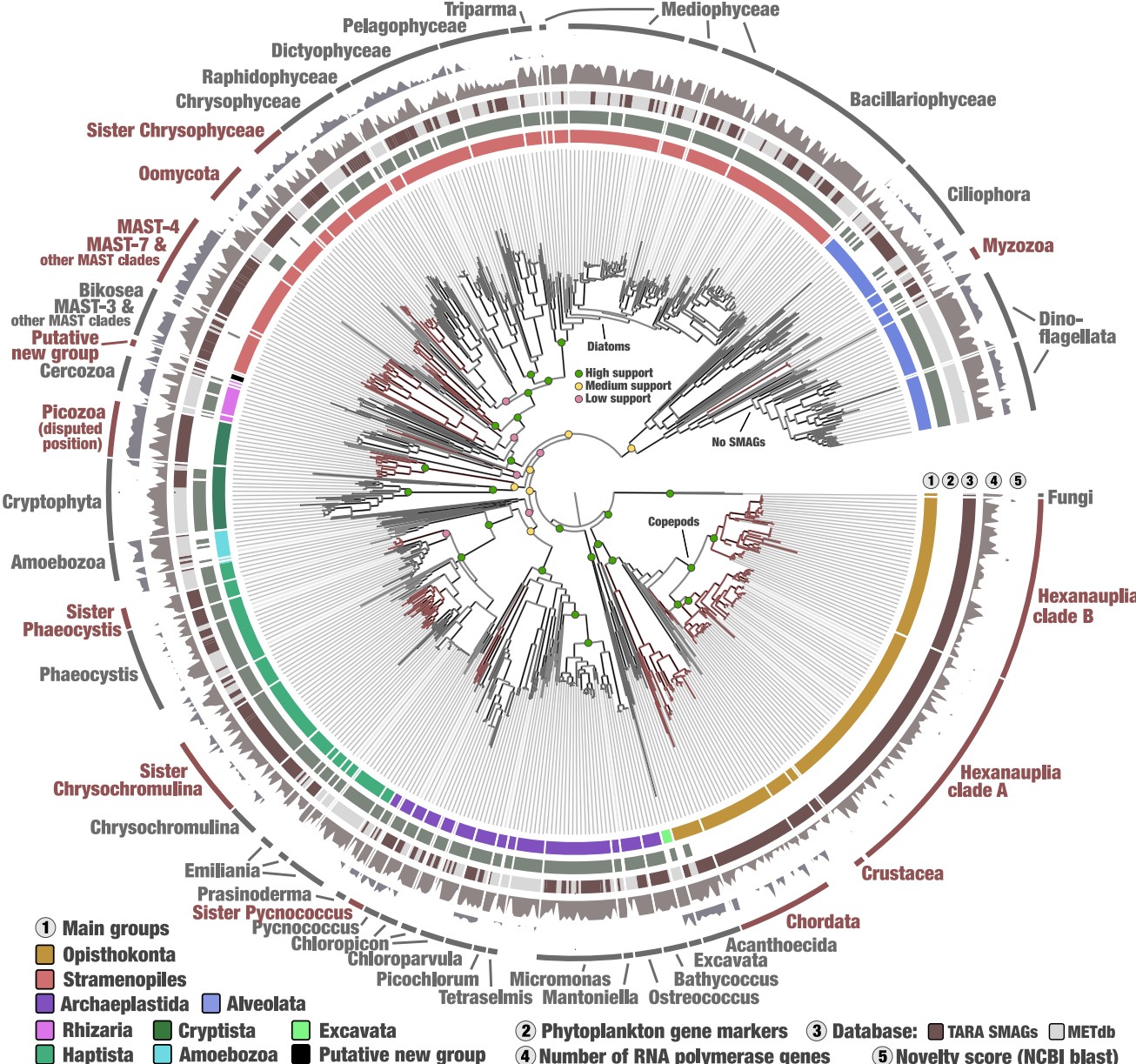

**Figure 2. Phylogenetic analysis of concatenated DNA-dependent RNA polymerase protein sequences from eukaryotic plankton**

The maximum-likelihood phylogenetic tree of the concatenated two largest subunits from the three DNA-dependent RNA polymerases (six genes in total) included *Tara* Oceans MAGs and SAGs and METdb transcriptomes and was generated using a total of 7,243 sites in the alignment and LG + F + R10 model; Opisthokonta was used as the outgroup. Supports for selected clades are displayed. Phylogenetic supports were considered high (aLRT $\geq$ 80 and UFBoot $\geq$ 95), medium (aLRT $\geq$ 80 or UFBoot $\geq$ 95), or low (aLRT < 80 and UFBoot < 95) (STAR Methods). The tree was decorated with additional layers using the anvi'o interface. The novelty score layer (STAR Methods) was set with a minimum of 30 (i.e., 70% similarity) and a maximum of 60 (i.e., 40% similarity). Branches and names in red correspond to main lineages lacking representatives in METdb.

reference proteins and phylogenetic inferences (STAR Methods and supplemental information). BLAST results provided a novelty score for each of them (STAR Methods and Table S3), expanding the scope of our analysis to eukaryotic genomes stored in NCBI as of August 2020. Our final phylogenetic analysis included 416 reference transcriptomes and 576 environmental MAGs and SAGs that contained at least one of the six marker

genes (Figure 2). The concatenated DNA-dependent RNA polymerase protein sequences effectively reconstructed a coherent tree of eukaryotic life, comparable to previous large-scale phylogenetic analyses based on other gene markers,[46] and to a complementary BUSCO-centric phylogenomic analysis using protein sequences corresponding to hundreds of smaller gene markers (Figure S2). As a noticeable difference, the Haptista

were most closely related to Archaeplastida, while Cryptista included the phylum Picozoa and was most closely related to the TSAR supergroup (Telonemia not represented here, Stramenopiles, Alveolata, and Rhizaria), albeit with weaker supports. This view of the eukaryotic tree of life using a previously under-exploited universal marker is by no means conclusive by itself but contributes to ongoing efforts to understand deep evolutionary relationships among eukaryotes while providing an effective framework to assess the phylogenetic positions of a large number of the *Tara* Oceans MAGs and SAGs.

Among small planktonic animals, the *Tara* Oceans MAGs recovered one lineage of Chordata related to the Oikopleuridae family, and Crustacea including a wide range of copepods (Figure 2; Table S3). Copepods dominate large size fractions of plankton[8] and represent some of the most abundant animals on the planet.[47,48] They actively feed on unicellular plankton and are a significant food source for larger animals such as fish, thus representing a key trophic link within the global carbon cycle.[49] For now, fewer than ten copepod genomes have been characterized by isolates.[50,51] The additional 8.4 Gbp of genomic material unveiled herein is split into 217 MAGs, and themselves organized into two main phylogenetic clusters that we dubbed marine Hexanauplia clades A and B. The two clades considerably expanded the known genomic diversity of copepods, albeit clade B was linked to few reference genomes (Figure S3). These clades were equally abundant and detected in all oceanic regions. Copepod MAGs typically had broad geographic distributions, being detected on average in 25% of the globally distributed *Tara* Oceans stations. In comparison, Opisthokonta MAGs affiliated to Chordata and Choanoflagellatea (Acanthoecida) were, on average, detected in less than 10% of sampling sites.

Generally occurring in smaller size fractions, MAGs and SAGs corresponding to unicellular eukaryotes considerably expanded our genomic knowledge of known genera within Alveolata, Archaeplastida, Haptista, and Stramenopiles (Figure 2; Table S3). Just within the diatoms for instance (Stramenopiles), MAGs were reconstructed for *Fragilariopsis* (n = 5), *Pseudo-nitzschia* (n = 7), *Chaetoceros* (n = 11), *Thalassiosira* (n = 5), and seven other genera (including the intriguing >1-Gbp-long genome of a blooming *O. weissflogii species*), all of which are known to contribute significantly to photosynthesis in the sunlit ocean.[52,53] Among the Archaeplastida, genome-wide average nucleotide identities and distribution patterns indicated that the large majority of MAGs correspond to distinct populations, many of which have not been characterized by means of culture genomics. Especially, we characterized the genomic content of at least 16 *Micromonas* populations (Figure S4), 11 *Chloropicon* populations (Figure S5), and five *Bathycoccus* populations (Figure S6). Beyond this genomic expansion of known planktonic genera, MAGs and SAGs covered various lineages lacking representatives in METdb. These included (1) Picozoa as a sister clade to Cryptista (SAGs from this phylum were recently linked to the Archaeplastida using different gene markers and databases[54]), to the class Chrysophyceae, and the genera *Phaeocystis* and *Pycnococcus*, (2) basal lineages of Oomycota within Stramenopiles and Myzozoa within Alveolata, (3) multiple branches within the MAST lineages[55] (Figure S7), (4) and a small cluster possibly

at the root of Rhizaria we dubbed "putative new group" (Figure S8). The novelty score of individual DNA-dependent RNA polymerase genes was supportive of the topology of the tree. Significantly, diverse MAST lineages, Picozoa, and the putative new group all displayed a deep branching distance from cultures and a high novelty score. In addition, the BUSCO-centric phylogenomic analysis placed the "putative new group" at the root of Haptista (Figure S2), supporting its high novelty while stressing the difficulty placing it accurately in the eukaryotic tree of life. In addition, this alternative phylogenomic analysis confirmed placement for the sister clade to *Phaeocystis* but not for the sister clade to *Pycnococcus*, placing it instead as a stand-alone lineage distinct from any Archaeplastida lineages represented by the MAGs, SAGs, and METdb. While different gene markers might provide slightly different evolutionary trends, a well-known phylogenetic phenomenon, here our two approaches concur when it comes to emphasizing the genomic novelty of the MAGs and SAGs compared with culture references.

One of the most conspicuous lineages lacking any MAGs and SAGs was the Dinoflagellata, a prominent and extremely diverse phylum in small and large eukaryotic size fractions of *Tara* Oceans.[8] These organisms harbor very large and complex genomes[56] that likely require much deeper sequencing efforts to be recovered by genome-resolved metagenomics. Besides, many other important lineages are also missing in MAGs and SAGs (e.g., within Radiolaria and Excavata), possibly due to a lack of abundant populations despite their diversity.

## A complex interplay between the evolution and functioning of marine eukaryotes

MAGs and SAGs provided a broad genomic assessment of the eukaryotic tree of life within the sunlit ocean by covering a wide range of marine plankton eukaryotes distantly related to cultures but abundant in the open ocean. Thus, the resource provided an opportunity to explore the interplay between the phylogenetic signal and functional repertoire of eukaryotic plankton with genomics. With EggNOG,[57–59] we identified orthologous groups corresponding to known (n = 15,870) and unknown functions (n = 12,567, orthologous groups with no assigned function at http://eggnog5.embl.de/) for 4.7 million genes (nearly 50% of the genes; STAR Methods). Among them, functional redundancy (i.e., a function detected multiple times in the same MAG or SAG) encompassed 46.6%–96.8% of the gene repertoires (average of 75.2% of functionally redundant genes). We then used these gene annotations to classify the MAGs and SAGs based on their functional profiles (Table S6). Our hierarchical clustering analysis using Euclidean distance and Ward linkage (an approach to organize genomes based on pangenomic traits[60]) first split the MAGs and SAGs into small animals (Chordata, Crustacea, copepods) and putative unicellular eukaryotes (Figure 3). Fine-grained functional clusters exhibited a highly coherent taxonomy within the unicellular eukaryotes. For instance, MAGs affiliated to the coccolithophore *Emiliana* (completion ranging from 7.8% to 32.2%), Dictyochophaceae family (completion ranging from 8.6% to 76.9%), and the sister clade to *Phaeocystis* (completion ranging from 18.4% to 60.4%) formed distinct clusters. The phylum Picozoa (completion ranging from 1.6% to 75.7%) was also confined to a single cluster that could be explained partly

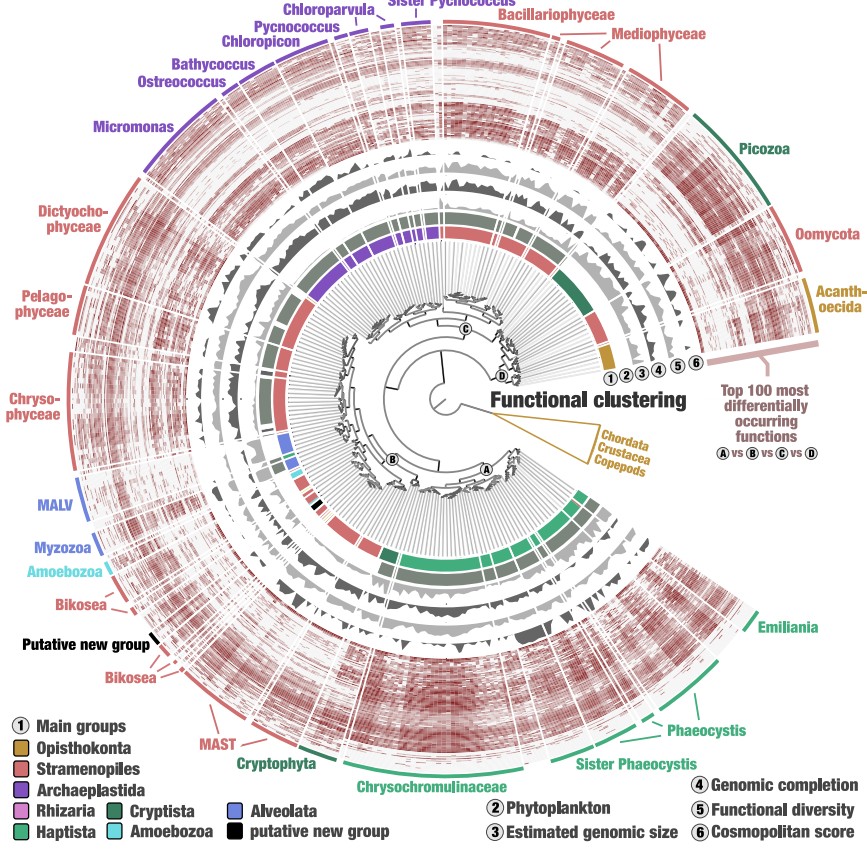

**Figure 3. The genomic functional landscape of unicellular eukaryotes in the sunlit ocean**

The figure displays a hierarchical clustering (Euclidean distance with Ward's linkage) of 681 MAGs and SAGs based on the occurrence of ~28,000 functions identified with EggNOG,[57–59] rooted with small animals (Chordata, Crustacea, and copepods) and decorated with layers of information using the anvi'o interactive interface. Layers include the occurrence in log 10 of 100 functions with lowest p value when performing Welch's ANOVA between the functional groups A, B, C and D (see nodes in the tree). Removed from the analysis were Ciliophora MAGs (gene calling is problematic for this lineage), two less complete MAGs affiliated to Opisthokonta, and functions occurring more than 500 times in the gigabase-scale MAG and linked to retrotransposons connecting otherwise unrelated MAGs and SAGs.

and Archaeplastida (small genomes) as sister clusters. This finding likely reflects that diatoms are the only group with an obligatory photoautotrophic lifestyle within the Stramenopiles, like the Archaeplastida. Finally, Group D encompassed three distantly related lineages of heterotrophs (those systematically lacked gene markers for photosynthesis) exhibiting rather large genomes: Oomycota, Acanthoecida choanoflagellates, and Picozoa. Those four functional groups have similar amounts of detected functions and contained both cosmopolite and rarely detected MAGs and SAGs across the *Tara* Oceans stations. While attempts to classify marine eukaryotes based on genomic functional traits have been made in the past (e.g., using a few SAGs[61]), our resource therefore provided a broad enough spectrum of genomic material for a first genome-wide functional classification of abundant lineages of unicellular eukaryotic plankton in the upper layer of the ocean.

A total of 2,588 known and 680 unknown functions covering 1.94 million genes (~40% of the annotated genes) were significantly differentially occurring between the four functional groups (Welch's ANOVA tests, p value <1.e$^{-05}$; Table S6). We displayed the occurrence of the 100 functions with lowest p values in the hierarchical clustering presented in Figure 3 to illustrate and help convey the strong signal between groups. However, more than 3,000 functions contributed to the basic partitioning of MAGs and SAGs. They cover all high-level functional categories identified in the 4.7 million genes with similar proportions (Figure S11), indicating that a wide range of functions related to information storage and processing, cellular processes and signaling, and metabolism contribute to the partitioning of the groups. As a notable difference, functions related to transcription (−50%) and RNA processing and modification (−47%) were less represented, while those related to carbohydrate transport and metabolism were enriched (+43%) in the

by a considerable radiation of genes related to dioxygenase activity (up to 644 genes). Most strikingly, the Archaeplastida MAGs not only clustered with respect to their genus-level taxonomy, but the organization of these clusters was highly coherent with their evolutionary relationships (see Figure 2), confirming not only the novelty of the putative sister clade to *Pycnococcus*, but also the sensitivity of our framework to draw the functional landscape of unicellular marine eukaryotes. Clearly, the important functional redundancy of MAGs and SAGs minimized the effect of genomic incompleteness in our efforts assessing the functional profile of unicellular marine eukaryotes.

Four major functional groups of unicellular eukaryotes emerged from the hierarchical clustering (Figure 3), which was perfectly recapitulated when incorporating the standard culture genomes matching to a MAG (Figure S9) and when clustering only the MAGs and SAGs >25% complete (Figure S10). Importantly, the taxonomic coherence observed in fine-grained clusters vanished when moving toward the root of these functional groups. Group A was an exception since it only covered the Haptista (including the highly cosmopolitan sister clade to *Phaeocystis*). Group B, on the other hand, encompassed a highly diverse and polyphyletic group of distantly related heterotrophic (e.g., MAST and MALV) and mixotrophic (e.g., Myzozoa and Cryptophyta) lineages of various genomic size, suggesting that broad genomic functional trends may not only be explained by the trophic mode of plankton. Group C was mostly photosynthetic and covered the diatoms (Stramenopiles of various genomic size)

thoecida choanoflagellates, and Picozoa. Those four functional groups have similar amounts of detected functions and contained both cosmopolite and rarely detected MAGs and SAGs across the *Tara* Oceans stations. While attempts to classify marine eukaryotes based on genomic functional traits have been made in the past (e.g., using a few SAGs[61]), our resource therefore provided a broad enough spectrum of genomic material for a first genome-wide functional classification of abundant lineages of unicellular eukaryotic plankton in the upper layer of the ocean.

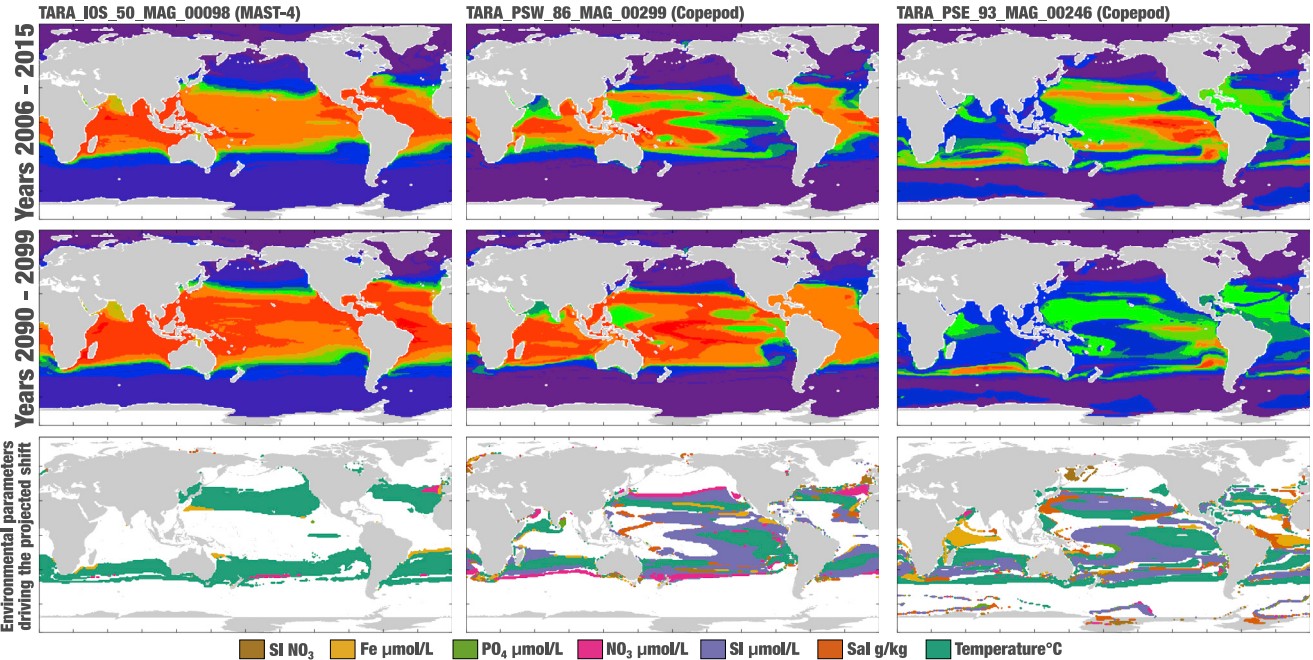

**Figure 4. World map distribution projections for three eukaryotic MAGs during the periods of 2006–15 and 2090–99**

The probability of presence ranges from 0 (purple) to 1 (red), with green corresponding to a probability of 0.5. The bottom row displays first-rank region-dependent environmental parameters driving the projected shifts of distribution (in regions where $|\Delta P| > 0.1$). Noticeably, projected decreases of silicate in equatorial regions drive 34% of the expansion of TARA_PSW_MAG_00,299 while driving 34% of the reduction of TARA_PSE_93_MAG_00,246, possibly reflecting different life strategies of these copepods (e.g., grazing). In contrast, the expansion of TARA_IOS_50_MAG_00,098 is mostly driven by temperature (74%).

differentially occurring functions. Interestingly, we noticed within Group C a scarcity of various functions otherwise occurring in high abundance among unicellular eukaryotes. These included functions related to ion channels (e.g., extracellular ligand-gated ion channel activity, intracellular chloride channel activity, magnesium ion transmembrane transporter activity, calcium ion transmembrane transport, calcium sodium antiporter activity) that may be linked to flagellar motility and the response to external stimuli,[62] reflecting the lifestyle of true autotrophs. Group D, on the other hand, had significant enrichment of various functions associated with carbohydrate transport and metabolism (e.g., alpha and beta-galactosidase activities, glycosyl hydrolase families, glycogen debranching enzyme, alpha-L-fucosidase), denoting a distinct carbon acquisition strategy. Overall, the properties of thousands of differentially occurring functions suggest that eukaryotic plankton's complex functional diversity is vastly intertwined within the tree of life, as inferred from phylogenies. This reflects the complex nature of the genomic structure and phenotypic evolution of organisms, which rarely fit their evolutionary relationships.

To this point, our analysis focused on the 4.4 million genes that were functionally annotated to EggNOG, which discarded more than half of the genes we identified in the MAGs and SAGs. Our current lack of understanding of many eukaryotic functional genes even within the scope of model organisms[63] can explain the limits of reference-based approaches to study the gene content of eukaryotic plankton. Thus, to gain further insights and overcome these limitations, we partitioned and categorized the eukaryotic gene content with AGNOSTOS.[64] AGNOSTOS

grouped 5.4 million genes in 424,837 groups of genes sharing remote homologies, adding 2.3 million genes left uncharacterized by the EggNOG annotation. AGNOSTOS applies a strict set of parameters for the grouping of genes discarding 575,053 genes by its quality controls and 4,264,489 genes in singletons. The integration of the EggNOG annotations into AGNOSTOS resulted in a combined dataset of 25,703 EggNOG orthologous groups (singletons and gene clusters) and 271,464 AGNOSTOS groups of genes, encompassing 6.4 million genes, 45% more genes that the original dataset (STAR Methods). The genome-wide functional classification of MAGs and SAGs based on this extended set of genes supported most trends previously observed with EggNOG annotation alone (Figure S12; Table S7), reinforcing our observations. But most interestingly, classification based solely on 23,674 newly identified groups of genes of unknown function (Table S8; a total of 1.3 million genes discarded by EggNOG) were also supportive of the overall trends, including notable links between diatoms and green algae and between Picozoa and Acanthoecida (Figure S13). Thus, we identified a functional repertoire convergence of distantly related eukaryotic plankton lineages in both the known and unknown coding sequence space, the latter representing a substantial amount of biologically relevant gene diversity.

## Niche and biogeography of individual eukaryotic populations

Besides insights into organismal evolution and genomic functions, the MAGs and SAGs provided an opportunity to evaluate

the present and future geographical distribution of eukaryotic planktonic populations (close to species-level resolution) using the genome-wide metagenomic read recruitments. Here, we determined the niche characteristics (e.g., temperature range) of 374 MAGs and SAGs (~50% of the resource) detected in at least five stations (Table S9) and used climate models to project world map distributions (http://end.mio.osupytheas.fr/Ecological_Niche_database/) based on climatologies for the periods of 2006–15 and 2090–99[24] (STAR Methods and supplemental information).

Each of these MAGs and SAGs was estimated to occur in a surface averaging 42 and 39 million $km^2$ for the first and second period, respectively, corresponding to ~12% of the surface of the ocean. Our data suggest that most eukaryotic populations in the database will remain widespread for decades to come. However, many changes in biogeography are projected to occur. For instance, the most widespread population in the first period (a MAST MAG) would still be ranked first at the end of the century but with a surface area increasing from 37% to 46% (Figure 4), a gain of 28 million $km^2$ corresponding to the surface of North America. Its expansion from the tropics toward more temperate oceanic regions regardless of longitude is mostly explained by temperature and reflects the expansion of tropical niches due to global warming, echoing recent predictions made with amplicon surveys and imaging data.[65] As an extreme case, the MAG benefiting the most between the two periods (a copepod) could experience a gain of 55 million $km^2$ (Figure 4), more than the surface of Asia and Europe combined. On the other hand, the MAG losing most ground (also a copepod) could undergo a decrease of 47 million $km^2$. Projected changes in these two examples correlated with various variables (including a notable contribution of silicate), an important reminder that temperature alone cannot explain plankton's biogeography in the ocean. Our integration of genomics, metagenomics, and climate models provided the resolution needed to project individual eukaryotic population niche trajectories in the sunlit ocean.

### Limitations of the study
Genome-resolved metagenomics applied to the considerable environmental DNA sequencing legacy of the *Tara* Oceans large cellular size fractions proved effective at complementing our culture portfolio of marine eukaryotes. Nevertheless, the approach failed to cover lineages (1) containing very large genomes (e.g., the Dinoflagelates[56]), (2) only found in low abundance, (3) or found to be abundant but with unusually high levels of microdiversity, challenging metagenomic assemblies (e.g., the prominent *Pelagomonas* genus[66] for which we only recovered high latitude MAG representatives). Deeper sequencing efforts coupled with long read sequencing technologies will likely overcome many of these limitations in years to come.

Our functional clustering of marine eukaryotes took advantage of a wide range of genomes manually characterized with the platform anvi'o, and also considered numerous gene clusters of unknown function using the AGNOSTOS framework. However, this methodology also contains noticeable limitations. For instance, clustering methodologies can influence the observed trends. Furthermore, integration of additional taxonomic groups that

currently lack genomic characterizations might impact functional clustering, similar to what is often observed with phylogenomic analyses. Thus, we anticipate that follow-up investigations might identify functional clusters slightly differing from the four major groups we have identified, refining our understanding of the functional convergence of distantly related eukaryotic lineages identified in our study.

### CONCLUSION

Similar to recent advances that elucidated viral, bacterial, and archaeal lineages, microbiology is experiencing a shift from cultivation to metagenomics for the genomic characterization of marine eukaryotes *en masse*. Indeed, our culture-independent and manually curated genomic characterization of abundant unicellular eukaryotic populations and microscopic animals in the sunlit ocean covers a wide range of poorly characterized lineages from multiple trophic levels (e.g., copepods and their prey, mixotrophs, autotrophs, and parasites) and provided the first gigabase-scale metagenome-assembled genome. Our genome-resolved survey and parallel efforts by others[67,68] are not only different from past transcriptomic surveys of isolated marine organisms but also better represent eukaryotic plankton in the open photic ocean. They represent innovative steps toward using genomics to explore in concert the ecological and evolutionary underpinnings of environmentally relevant eukaryotic organisms, using metagenomics to fill critical gaps in our remarkable culture porfolio.[21]

Phylogenetic gene markers such as the DNA-dependent RNA polymerases (the basis of our phylogenetic analysis) provide a critical understanding of the origin of eukaryotic lineages and allowed us to place most environmental genomes in a comprehensible evolutionary framework. However, this framework is based on sequence variations within core genes that in theory are inherited from the last eukaryotic common ancestor representing the vertical evolution of eukaryotes, disconnected from the structure of genomes. As such, it does not recapitulate the functional evolutionary journey of plankton, as demonstrated in our genome-wide functional classification of unicellular eukaryotes in both the known and unknown coding sequence space. The dichotomy between phylogeny and function was already well described with morphological and other phenotypic traits and could be explained in part by secondary endosymbiosis events that have spread plastids and genes for their photosynthetic capabilities across the eukaryotic tree of life.[69–72] Here we moved beyond morphological inferences and disentangled the phylogeny of gene markers and broad genomic functional repertoire of a comprehensive collection of marine eukaryotic lineages. We identified four major genomic functional groups of unicellular eukaryotes made of distantly related lineages. The Stramenopiles proved particularly effective in terms of genomic functional diversification, possibly explaining part of their remarkable success in this biome.[8,73]

The topology of phylogenetic trees compared to the functional clustering of a wide range of eukaryotic lineages has revealed contrasting evolutionary journeys for widely scrutinized gene markers of evolution and less studied genomic functions of plankton. The apparent functional convergence of distantly

related lineages that coexisted in the same biome for millions of years could not be explained by either a vertical evolutionary history of unicellular eukaryotes nor their trophic modes (phytoplankton versus heterotrophs), shedding new lights into the complex functional dynamics of plankton over evolutionary time scales. Convergent evolution is a well-known phenomenon of independent origin of biological traits such as molecules and behaviors[74,75] that has been observed in the morphology of microbial eukaryotes[76] and is often driven by common selective pressures within similar environmental conditions. However, an independent origin of similar functional profiles is not the only possible explanation for organisms sharing the same habitat. Indeed, one could wonder if lateral gene transfers between eukaryotes[77,78] have played a central role in these processes, as previously observed between eukaryotic plant pathogens[79] or grasses.[80] As a case in point, secondary endosymbiosis events are known to have resulted in massive gene transfers between endosymbionts and their hosts in the oceans.[69,70] In particular, these events involved transfers of genes from green algae to diatoms,[81] two lineages clustering together in our genomic functional classification of eukaryotic plankton. However, lineages sharing the same secondary endosymbiotic history did not always fall in the same functional group. This was the case for diatoms, Haptista, and Cryptista that have different functional trends yet originate from a common ancestor that likely acquired its plastid from red and green algae.[69,70,82] Surveying phylogenetic trends for functions derived from the ~10 million genes identified here will likely contribute to new insights regarding the extent of lateral gene transfers between eukaryotes,[83,84] the independent emergence of functional traits (convergent evolution), as well as functional losses between lineages,[85] that altogether might have driven the functional convergences of distantly related eukaryotic lineages abundant in the sunlit ocean.

Regardless of the mechanisms involved, the functional repertoire convergences we observed likely highlight primary organismal functioning, which have fundamental impacts on plankton ecology, and their functions within marine ecosystems and biogeochemical cycles. Thus, the apparent dichotomy between phylogenies (a vertical evolutionary framework) and genome-wide functional repertoires (genome structure evolution) depicted here should be viewed as a fundamental attribute of marine unicellular eukaryotes that we suggest warrants a new rationale for studying the structure and state of plankton, a rationale also based on present-day genomic functions rather than phylogenetic and morphological surveys alone.

## CONSORTIA

Shinichi Sunagawa, Silvia G. Acinas, Peer Bork, Eric Karsenti, Chris Bowler, Christian Sardet, Lars Stemmann, Colomban de Vargas, Patrick Wincker, Magali Lescot, Marcel Babin, Gabriel Gorsky, Nigel Grimsley, Lionel Guidi, Pascal Hingamp, Olivier Jaillon, Stefanie Kandels, Daniele Iudicone, Hiroyuki Ogata, Stéphane Pesant, Matthew B. Sullivan, Fabrice Not, Lee Karp-Boss, Emmanuel Boss, Guy Cochrane, Michael Follows, Nicole Poulton, Jeroen Raes, Mike Sieracki, and Sabrina Speich.

## STAR★METHODS

Detailed methods are provided in the online version of this paper and include the following:

- KEY RESOURCES TABLE
- RESOURCE AVAILABILITY
  - Lead contact
  - Materials availability
  - Data and code availability
- METHOD DETAILS
  - *Tara* Oceans metagenomes
  - Genome-resolved metagenomics
  - A first gigabase scale eukaryotic MAG
  - MAGs from the 0.2–3 μm size fraction
  - Single-cell genomics
  - Characterization of a non-redundant database of MAGs and SAGs
  - Taxonomical inference of MAGs and SAGs
  - Protein coding genes
  - Protein-coding genes for the MAGs
  - Protein coding genes for the SAGs
  - BUSCO completion scores for protein-coding genes in MAGs and SAGs
  - Biogeography of MAGs and SAGs
  - Identifying the environmental niche of MAGs and SAGs
  - Cosmopolitan score
  - A database of manually curated DNA-dependent RNA polymerase genes
  - Novelty score for the DNA-dependent RNA polymerase genes
  - Phylogenetic analyses of MAGs and SAGs
  - EggNOG functional inference of MAGs and SAGs
  - Eukaryotic MAGs and SAGs integration in the AGNOSTOS-DB
  - AGNOSTOS functional aggregation inference
  - Functional clustering of MAGs and SAGs
- QUANTIFICATION AND STATISTICAL ANALYSIS
  - Differential occurrence of functions

### SUPPLEMENTAL INFORMATION

### ACKNOWLEDGMENTS

Our survey was made possible by two scientific endeavors: the sampling and sequencing efforts by the *Tara* Oceans Project and the bioinformatics and visualization capabilities afforded by anvi'o. We are indebted to all who contributed to these efforts, as well as other open-source bioinformatics tools for their commitment to transparency and openness. *Tara* Oceans (which includes the *Tara* Oceans and *Tara* Oceans Polar Circle expeditions) would not exist without the leadership of the *Tara* Ocean Foundation and the continuous support of 23 institutes (https://oceans.taraexpeditions.org/). We thank the commitment of the following people and sponsors who made this singular expedition possible: CNRS (in particular Groupement de Recherche GDR3280 and the Research Federation for the Study of Global Ocean Systems Ecology and Evolution FR2022/Tara GOSEE), the European Molecular Biology Laboratory (EMBL), Genoscope/CEA, the French Ministry of Research and the French Governement 'Investissement d'Avenir'

programs Oceanomics (ANR-11-BTBR-0008), FRANCE GENOMIQUE (ANR-10-INBS-09), ATIGE Genopole postdoctoral fellowship, HYDROGEN/ANR-14-CE23-0001, MEMO LIFE (ANR-10-LABX-54), PSL Research University (ANR-11-IDEX-0001-02) and EMBRC-France (ANR-10-INBS-02), Fund for Scientific Research—Flanders, VIB, Stazione Zoologica Anton Dohrn, UNIMIB, ANR (projects ALGALVIRUS ANR-17-CE02- 0012, PHYTBACK/ANR-2010-1709-01, POSEIDON/ANR-09-BLAN-0348, PROMETHEUS/ANR-09-PCS-GENM-217, TARA-GIRUS/ANR-09-PCS-GENM-218), EU FP7 (MicroB3/No. 287589, IHMS/HEALTH-F4-2010-261376), Genopole, CEA DRF Impulsion program, OCEANOMICS (project no. ANR-11-BTBR-0008), ERC Advanced Award Diatomic (grant agreement No 835067) to CB. The authors also thank Agnès B. and E. Bourgois, the Prince Albert II de Monaco Foundation, the Veolia Foundation, the EDF Foundation, Region Bretagne, Lorient Agglomeration, Worldcourier, Illumina, Serge Ferrari, and the Fonds Francais pour l'Environnement Mondial for support and commitment. The global sampling effort was made possible by countless scientists and crew who performed sampling aboard the *Tara* from 2009 to 2013. The authors are also grateful to the countries that graciously granted sampling permission. Part of the computations were performed using the platine, titane, and curie HPC machine provided through GENCI grants (t2011076389, t2012076389, t2013036389, t2014036389, t2015036389, and t2016036389). We also thank Noan Le Bescot (TernogDesign) for artwork on figures.

## AUTHOR CONTRIBUTIONS

D.D.H., M.G., E.P., P.W., O.J., and T.O.D. conducted the study. T.O.D. and M.G. characterized the MAGs and SAGs, and RNA polymerase genes, respectively. D.D.H. (analysis of the ~10 million genes), M.G. (phylogenies), P.F. (climate models and world map projections), E.P. (METdb database, mapping results), and T.O.D. performed the primary analysis of the data. A.K., L.d.A., Q.C., and J.-M.A. assembled and annotated the single cell genomes and helped to process metagenomic assemblies. E.V., M.W., B.N., C.D.S., D.D.H., O.J., and J.-M.A. identified the eukaryotic genes in the MAG assemblies. A.F.G. and C.V. characterized the repertoire of functions in the unknown coding sequence space. T.O.D. wrote the manuscript with critical inputs from all the authors. This article is contribution number 132 of *Tara* Oceans.

## DECLARATION OF INTERESTS

The authors declare no competing interests.

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

# STAR★METHODS

## KEY RESOURCES TABLE

| REAGENT or RESOURCE | SOURCE | IDENTIFIER |
|---|---|---|
| **Deposited data** | | |
| Data generated in this study | This paper | https://www.genoscope.cns.fr/tara/ |
| Data needed for the Agnostos related analyses for this study | This paper | https://figshare.com/articles/dataset/Delmont_et_al_2022/19403531 |
| **Software and algorithms** | | |
| Anvi'o | Eren et al. 2021 | https://anvio.org/ |
| Agnostos | Vanni et al., 2021 | https://github.com/functional-dark-side/agnostos-wf |
| Custom codes required to perform analyses related to Agnostos in this study | This paper | https://zenodo.org/record/6379623 |

## RESOURCE AVAILABILITY

### Lead contact
Further information and requests for resources and analyses should be directed to and will be fulfilled by the lead contact, Tom O. Delmont (Tom.Delmont@genoscope.fr).

### Materials availability
This study did not generate new materials.

### Data and code availability
- All data our study generated are publicly available at http://www.genoscope.cns.fr/tara/. The link provides access to the 11 raw metagenomic co-assemblies, the FASTA files for 713 MAGs and SAGs, the ~10 million protein-coding sequences (nucleotides, amino acids and gff format), and the curated DNA-dependent RNA polymerase genes (MAGs and SAGs and METdb transcriptomes). This link also provides access to the supplemental figures and the Supplemental material. Finally, code development within anvi'o for the BUSCO single copy core genes is available at https://github.com/merenlab/anvio.
- Original code has been deposited at Zenodo and is publicly available. The accession number is listed in the key resources table.
- Any additional information required to reanalyze the data reported in this paper is available from the lead contact upon request.

## METHOD DETAILS

### *Tara* Oceans metagenomes
We analyzed a total of 939 *Tara Oceans* metagenomes available at the EBI under project PRJEB402 (https://www.ebi.ac.uk/ena/browser/view/PRJEB402). 265 of these metagenomes have been released through this study. Table S1 reports accession numbers and additional information (including the number of reads and environmental metadata) for each metagenome.

### Genome-resolved metagenomics
We organized the 798 metagenomes corresponding to size fractions ranging from 0.8 μm to 2 mm into 11 'metagenomic sets' based upon their geographic coordinates. We used those 0.28 trillion reads as inputs for 11 metagenomic co-assemblies using MEGAHIT[86] v1.1.1, and simplified the scaffold header names in the resulting assembly outputs using anvi'o[38,39] v.6.1 (available from http://merenlab.org/software/anvio). Co-assemblies yielded 78 million scaffolds longer than 1,000 nucleotides for a total volume of 150.7 Gbp. We performed a combination of automatic and manual binning on each co-assembly output, focusing only on the 11.9 million scaffolds longer than 2,500 nucleotides, which resulted in 837 manually curated eukaryotic metagenome-assembled genomes (MAGs) longer than 10 million nucleotides. Briefly, (1) anvi'o profiled the scaffolds using Prodigal[87] v2.6.3 with default parameters to identify an initial set of genes, and HMMER[88] v3.1b2 to detect genes matching to 83 single-copy core gene markers from BUSCO[89] (benchmarking is described in a dedicated blog post[90]), (2) we used a customized database including both NCBI's NT database and METdb to infer the taxonomy of genes with a Last Common Ancestor strategy[5] (results were imported as described in http://merenlab.org/2016/06/18/importing-taxonomy), (3) we mapped short reads from the metagenomic set to the scaffolds using BWA v0.7.15[91] (minimum identity of 95%) and stored the recruited reads as BAM files using samtools,[92] (4) anvi'o profiled each BAM file to estimate the coverage and detection statistics of each scaffold, and combined mapping profiles into a merged profile database

for each metagenomic set. We then clustered scaffolds with the automatic binning algorithm CONCOCT[93] by constraining the number of clusters (thereafter dubbed metabins) per metagenomic set to a number ranging from 50 to 400 depending on the set. Each metabin (n = 2,550, ~12 million scaffolds) was manually binned using the anvi'o interactive interface. The interface considers the sequence composition, differential coverage, GC-content, and taxonomic signal of each scaffold. Finally, we individually refined each eukaryotic MAG >10 Mbp as outlined in Delmont and Eren,[94] and renamed scaffolds they contained according to their MAG ID. Table S2 reports the genomic features (including completion and redundancy values) of the eukaryotic MAGs. For details on our protocol used for binning and curation of metabins, see Methods S1, Supplemental methods, Related to the STAR Methods.

### A first gigabase scale eukaryotic MAG

We performed targeted genome-resolved metagenomics to confirm the biological relevance and improve statistics of the single MAG longer than one Gbp with an additional co-assembly (five Southern Ocean metagenomes for which this MAG had average vertical coverage >1x) and by considering contigs longer than 1,000 nucleotides, leading to a gain of 181,8 million nucleotides. To our knowledge, we describe here the first successful characterization of a Gigabase-scale MAG (1.32 Gbp with 419,520 scaffolds), which we could identify using two distinct metagenomic co-assemblies.

### MAGs from the 0.2–3 μm size fraction

We incorporated into our database 20 eukaryotic MAGs longer than 10 million nucleotides previously characterized from the 0.2-3 μm size fraction,[26] providing a set of MAGs corresponding to eukaryotic cells ranging from 0.2 μm (picoeukaryotes) to 2 mm (small animals).

### Single-cell genomics

We used 158 eukaryotic single cells sorted by flow cytometry from seven *Tara* Oceans stations as input to perform genomic assemblies (up to 18 cells with identical 18S rRNA genes per assembly to optimize completion statistics, see Table S2), providing 34 single-cell genomes (SAGs) longer than 10 million nucleotides. Cell sorting, DNA amplification, sequencing and assembly were performed as described elsewhere.[18] In addition, manual curation was performed using sequence composition and differential coverage across 100 metagenomes in which the SAGs were most detected, following the methodology described in the genome-resolved metagenomics section. For SAGs with no detection in *Tara* Oceans metagenomes, only sequence composition and taxonomical signal could be used, limiting this curation effort's scope. Notably, manual curation of SAGs using the genome-resolved metagenomic workflow turned out to be highly valuable, leading to the removal of more than one hundred thousand scaffolds for a total volume of 193.1 million nucleotides. This metagenomic-guided decontamination effort contributes to previous efforts characterizing eukaryotic SAGs from the same cell sorting material[18,61,95–97] and provides new marine eukaryotic guidelines for SAGs. For details on our protocol used for curation of eukaryotic SAGs, see Methods S1, Supplemental methods, Related to the STAR Methods.

### Characterization of a non-redundant database of MAGs and SAGs

We determined the average nucleotide identity (ANI) of each pair of MAGs and SAGs using the dnadiff tool from the MUMmer package[98] v.4.0b2. MAGs and SAGs were considered redundant when their ANI was >98% (minimum alignment of >25% of the smaller MAG or SAG in each comparison). We then selected the longest MAG or SAG to represent a group of redundant MAGs and SAGs. This analysis provided a non-redundant genomic database of 713 MAGs and SAGs.

### Taxonomical inference of MAGs and SAGs

We manually determined the taxonomy of MAGs and SAGs using a combination of approaches: (1) taxonomical signal from the initial gene calling (Prodigal), (2) phylogenetic approaches using the RNA polymerase genes and METdb, (3) ANI within the MAGs and SAGs and between MAGs and SAGs and METdb, (4) local blasts using BUSCO gene markers, (5) and lastly the functional clustering of MAGs and SAGs to gain knowledge into very few MAGs and SAGs lacking gene markers and ANI signal. In addition, Picozoa SAGs[54] were used to identify MAGs from this phylum lacking representatives in METdb. For details on METdb, see Methods S1, Supplemental methods, Related to the STAR Methods.

### Protein coding genes

Protein coding genes for the MAGs and SAGs were characterized using three complementary approaches: protein alignments using reference databases, metatranscriptomic mapping from *Tara* Oceans and *ab-initio* gene predictions. While the overall framework was highly similar for MAGs and SAGs, the methodology slightly differed to take the best advantage of those two databases when they were processed (see the two following sections).

### Protein-coding genes for the MAGs
#### Protein alignments

Since the alignment of a large protein database on all the MAG assemblies is time greedy, we first detected the potential proteins of Uniref. 90 + METdb that could be aligned to the assembly by using MetaEuk[99] with default parameters. This subset of proteins was aligned using BLAT with default parameters, which localized each protein on the MAG assembly. The exon/intron structure was

refined using genewise[100] with default parameters to detect splice sites accurately. Each MAG's GeneWise alignments were converted into a standard GFF file and given as input to gmove.

### Metatranscriptomic mapping from Tara Oceans

A total of 905 individual *Tara* Oceans metatranscriptomic assemblies (mostly from large planktonic size fractions) were aligned on each MAG assembly using Minimap2[101] (version 2.15-r905) with the "-ax splice" flag. BAM files were filtered as follows: low complexity alignments were removed and only alignments covering at least 80% of a given metatranscriptomic contig with at least 95% of identity were retained. The BAM files were converted into a standard GFF file and given as input to gmove.

### Ab-initio gene predictions

A first gene prediction for each MAG was performed using gmove and the GFF file generated from metatranscriptomic alignments. From these preliminary gene models, 300 gene models with a start and a stop codon were randomly selected and used to train AUGUSTUS[102] (version 3.3.3). A second time, AUGUSTUS was launched on each MAG assembly using the dedicated calibration file, and output files were converted into standard GFF files and given as input to gmove. Each individual line of evidence was used as input for gmove (http://www.genoscope.cns.fr/externe/gmove/) with default parameters to generate the final protein-coding genes annotations.

### Protein coding genes for the SAGs

#### Protein alignments

The Uniref90 + METdb database of proteins was aligned using BLAT[103] with default parameters, which localized protein on each SAG assembly. The exon/intron structure was refined using GeneWise[100] and default parameters to detect splice sites accurately. The GeneWise alignments of each SAG were converted into a standard GFF file and given as input to gmove.

#### Metatranscriptomic mapping from Tara Oceans

The 905 *Tara* Oceans metatranscriptomic individual fastq files were filtered with kfir (http://www.genoscope.cns.fr/kfir) using a k-mer approach to select only reads that shared 25-mer with the input SAG assembly. This subset of reads was aligned on the corresponding SAG assembly using STAR[104] (version 2.5.2.b) with default parameters. BAM files were filtered as follows: low complexity alignments were removed and only alignments covering at least 80% of the metatranscriptomic reads with at least 90% of identity were retained. Candidate introns and exons were extracted from the BAM files and given as input to gmorse.[105]

#### Ab-initio gene predictions

*Ab-initio* models were predicted using SNAP[106] (v2013-02-16) trained on complete protein matches and gmorse models, and output files were converted into standard GFF files and given as input to gmove. Each line of evidence was used as input for gmove (http://www.genoscope.cns.fr/externe/gmove/) with default parameters to generate the final protein-coding genes annotations.

### BUSCO completion scores for protein-coding genes in MAGs and SAGs

BUSCO[89] v.3.0.4 was used with the set of eukaryotic single-copy core gene markers (n = 255). Completion and redundancy (number of duplicated gene markers) of MAGs and SAGs were computed from this analysis.

### Biogeography of MAGs and SAGs

We performed a final mapping of all metagenomes to calculate the mean coverage and detection of the MAGs and SAGs (Table S5). Briefly, we used BWA v0.7.15 (minimum identity of 90%) and a FASTA file containing the 713 non-redundant MAGs and SAGs to recruit short reads from all 939 metagenomes. We considered MAGs and SAGs were detected in a given filter when >25% of their length was covered by reads to minimize non-specific read recruitments.[26] The number of recruited reads below this cut-off was set to 0 before determining vertical coverage and percent of recruited reads. Regarding the projection of mapped reads, if MAGs and SAGs were to be complete, we used BUSCO completion scores to project the number of mapped reads. Note that we preserved the actual number of mapped reads for the MAGs and SAGs with completion <10% to avoid substantial errors to be made in the projections.

### Identifying the environmental niche of MAGs and SAGs

Seven physicochemical parameters were used to define environmental niches: sea surface temperature (SST), salinity (Sal), dissolved silica (Si), nitrate ($NO_3$), phosphate ($PO_4$), iron (Fe), and a seasonality index of nitrate (SI $NO_3$). Except for Fe and SI NO3, these parameters were extracted from the gridded World Ocean Atlas 2013 (WOA13).[107] Climatological Fe fields were provided by the biogeochemical model PISCES-v2.[108] The seasonality index of nitrate was defined as the range of nitrate concentration in one grid cell divided by the maximum range encountered in WOA13 at the Tara sampling stations. All parameters were co-located with the corresponding stations and extracted at the month corresponding to the Tara sampling. To compensate for missing physicochemical samples in the Tara *in situ* dataset, climatological data (WOA) were favored. For details on the environmental niches, see Methods S1, Supplemental methods, Related to the STAR Methods.

### Cosmopolitan score

Using metagenomes from the Station subset 1 (n = 757), MAGs and SAGs were assigned a "cosmopolitan score" based on their detection across 119 stations. For details on metagenomic subsets, see Methods S1, Supplemental methods, Related to the STAR Methods.

### A database of manually curated DNA-dependent RNA polymerase genes

A eukaryotic dataset[109] was used to build HMM profiles for the two largest subunits of the DNA-dependent RNA polymerase (RNAP-a and RNAP-b). These two HMM profiles were incorporated within the anvi'o framework to identify RNAP-a and RNAP-b genes (Prodigal[87] annotation) in the MAGs and SAGs and METdb transcriptomes. Alignments, phylogenetic trees and blast results were used to organize and manually curate those genes. Finally, we removed sequences shorter than 200 amino-acids, providing a final collection of DNA-dependent RNA polymerase genes for the MAGs and SAGs (n = 2,150) and METdb (n = 2,032) with no duplicates. For details on this protocol, see Methods S1, Supplemental methods, Related to the STAR Methods.

### Novelty score for the DNA-dependent RNA polymerase genes

We compared both the RNA-Pol A and RNA-Pol B peptides sequences identified in MAGs and SAGs and MetDB to the nr database (retrieved on October 25, 2019) using blastp, as implemented in blast+[110] v.2.10.0 (e-value of $1e^{-10}$). We kept the best hit and considered it as the closest sequence present in the public database. For each MAG and SAG, we computed the average percent identity across RNA polymerase genes (up to six genes) and defined the novelty score by subtracting this number from 100. For example, with an average percent identity of 64%, the novelty score would be 36%.

### Phylogenetic analyses of MAGs and SAGs

The protein sequences included for the phylogenetic analyses (either the **DNA-dependent RNA polymerase genes** we recovered manually or the **BUSCO set of 255 eukaryotic single-copy core gene markers** we recovered automatically from the ∼10 million protein coding genes) were aligned with MAFFT[111] v.764 and the FFT-NS-i algorithm with default parameters. Sites with more than 50% of gaps were trimmed using Goalign v0.3.0-alpha5 (http://www.github.com/evolbioinfo/goalign). The phylogenetic trees were reconstructed with IQ-TREE[112] v1.6.12, and the model of evolution was estimated with the ModelFinder[113] Plus option: for the concatenated tree, the LG + F + R10 model was selected. Supports were computed from 1,000 replicates for the Shimodaira-Hasegawa (SH)-like approximation likelihood ratio (aLRT)[114] and ultrafast bootstrap approximation (UFBoot).[115] As per IQ-TREE manual, we deemed the supports good when SH-aLRT ≥ 80% and UFBoot ≥ 95%. Anvi'o v.6.1 was used to visualize and root the phylogenetic trees.

### EggNOG functional inference of MAGs and SAGs

We performed the functional annotation of protein-coding genes using the EggNog-mapper[58,59] v2.0.0 and the EggNog5 database.[57] We used Diamond[116] v0.9.25 to align proteins to the database. We refined the functional annotations by selecting the orthologous group within the lowest taxonomic level predicted by EggNog-mapper.

### Eukaryotic MAGs and SAGs integration in the AGNOSTOS-DB

We used the AGNOSTOS workflow to integrate the protein coding genes predicted from the MAGs and SAGs into a variant of the AGNOSTOS-DB that contains 1,829 metagenomes from the marine and human microbiomes, 28,941 archaeal and bacterial genomes from the Genome Taxonomy Database (GTDB) and 3,243 nucleocytoplasmic large DNA viruses (NCLDV) metagenome assembled genomes (MAGs).[64]

### AGNOSTOS functional aggregation inference

AGNOSTOS partitioned protein coding genes from the MAGs and SAGs in groups connected by remote homologies, and categorized those groups as members of the known or unknown coding sequence space based on the workflow described in Vanni et al. 2020.[64] To combine the results from AGNOSTOS and the EggNOG classification we identified those groups of genes in the known space that contain genes annotated with an EggNOG and we inferred a consensus annotation using a quorum majority voting approach. AGNOSTOS produces groups of genes with low functional entropy in terms of EggNOG annotations as shown in Vanni et al. 2020[64] allowing us to combine both sources of information. We merged the groups of genes that shared the same consensus EggNOG annotations and we integrated them with the rest of AGNOSTOS groups of genes, mostly representing the unknown coding sequence space. Finally, we excluded groups of genes occurring in less than 2% of the MAGs and SAGs.

### Functional clustering of MAGs and SAGs

We used anvi'o to cluster MAGs and SAGs as a function of their functional profile (Euclidean distance with ward's linkage), and the anvi'o interactive interface to visualize the hierarchical clustering in the context of complementary information.

### QUANTIFICATION AND STATISTICAL ANALYSIS

### Differential occurrence of functions

We performed a Welch's ANOVA test followed by a Games-Howell test for significant ANOVA comparisons to identify EggNOG functions occurring differentially between functional groups of MAGs and SAGs. All statistics were generated in R 3.5.3. Results are available in the Table S6.

