## [Document S2. Transparent peer review record for Delmont et al · Cell Genomics]

Functional repertoire convergence of distantly related eukaryotic plankton lineages abundant in the sunlit ocean

Tom O. Delmont*^{1,2}, Morgan Gaia^{1,2}, Damien D. Hinsinger^{1,2}, Paul Fremont^{1,2}, Chiara Vanni³, Antonio Fernandez Guerra^{3,4}, A. Murat Eren^{5,6}, Artem Kourlaiev^{1,2}, Leo d'Agata^{1,2}, Quentin Clayssen^{1,2}, Emilie Villar¹, Karine Labadie^{1,2}, Corinne Cruaud^{1,2}, Julie Poulain^{1,2}, Corinne Da Silva^{1,2}, Marc Wessner^{1,2}, Benjamin Noel^{1,2}, Jean-Marc Aury^{1,2}, *Tara* Oceans Coordinators, Colombar de Vargas^{2,7}, Chris Bowler^{2,8}, Eric Karsenti^{2,7,9}, Eric Pelletier^{1,2}, Patrick Wincker^{1,2} and Olivier Jaillon^{1,2}

Summary

Initial submission: Received : Feb 16 2021

Scientific editors: Orli Bahcall and Rita Gemayel

First round of review: Number of reviewers: 2
Revision invited : Sep 10 2021
Revision received : Dec 10 2021

Second round of review: Number of reviewers: 2
Accepted : Apr 1 2022

Data freely available: yes

Code freely available: yes

This transparent peer review record is not systematically proofread, type-set, or edited. Special characters, formatting, and equations may fail to render properly. Standard procedural text within the editor's letters has been deleted for the sake of brevity, but all official correspondence specific to the manuscript has been preserved.

Referees' reports, first round of review

Reviewer #1 (Comments to authors)

Delmont et al. report the recovery of over 700 environmental eukaryotic genome bins from a combination of Tara Oceans shotgun metagenomic sequencing and single cell genomics samples. The binned genomes greatly expand the sequenced diversity of ocean eukaryotes, particularly of copepods. Further, these genome bins provide important references for existing Tara Oceans eukaryotic transcriptomic and gene centric approaches, allowing for analysis of the functional repertoire of organisms represented in these datasets. Functional classification and clustering of the eukaryotic bins revealed four primary functional groups, composed of distantly related lineages and suggesting functional convergence, but is a result that may be unduly influenced by completeness of the individual bins. A highlight of this report is that it brings important new approaches for the manual curation and binning of eukaryotic genomes and provides a detailed methodology, a valuable resource for the community.

At its core, this paper addresses the functional and phylogenetic diversity of microbial eukaryotes in the oceans. The methods are modified from standard bacterial shotgun metagenomic methods to be applied to microbial eukaryotes in a novel way. The authors report important findings, however there are areas in need of improvement. In addition, the manuscript would benefit from more clear or expanded justification on why some choices were made, such as the inclusion of low completeness bins which are typically filtered, the choice of proteins for Figure 2, and the initial decision to use a co-assembly based approach.

Major points of criticism that should be addressed:

The bin completeness distribution should be clarified in the main text. Upon first reading, it was not clear the 700+ genome bins range widely in completeness (0% to 93.7% BUSCO completeness, Table S3), information that is needed to properly interpret the main findings of the manuscript. The authors use a minimum bin size cutoff (10 mbp, l. 118) for filtering eukaryotic bins of interest, but this alone is not sufficient to make the quality of the individual genomes interpretable as eukaryotic genome size is independent of gene content and thus, genome completeness.

L. 294 Given the wide range of genome completeness across included SMAGs, genome completeness is likely to influence clustering based on presence or absence of functional repertoire. As written, it is not clear this is controlled for in the current analysis. Do the same patterns of clustering (general structure of the four major groups) hold when excluding low completeness bins? Do the functional groups retain their structure when closely related cultured representatives are included?

L. 108 Large co-assemblies can assemble poorly due to introduced heterogeneity, and may explain some of the highly incomplete genomes. Has targeted reassembly of individual samples or smaller co-assemblies been attempted? If not, assembling and binning a subset of samples would be useful for comparison, particularly because this paper will serve as an important reference for methodological approaches and knowing the potential trade-offs and benefits of co-assembly compared to deep sequencing of individual samples will be beneficial.

Minor points:

L. 137 It is interesting there were bins with such low GC content given illumina sequencing bias. Are these bins typically low completeness or low coverage? It may be worth noting them separately

L. 166 Metagenomics has the potential to better represent environmental eukaryotic diversity than culturing, but mapping reads from the samples you assembled your genomes from and comparing to cultured genomes is not an unbiased way to make this argument. Are there representative ocean samples from which you didn't assemble that could be used for mapping?

General statistics on cultured ocean eukaryotic genome structure, such as genome size, GC, etc, would be useful for context and comparison to the SMAGs.

L. 261 How large of a role do the authors think extraction played in the types of genomes recovered? Could this partially explain the missing Dinoflagellata?

Figure 2: It is difficult to spot the putative new group in the tree.

Supplemental figure S1: branch supports and additional tracks on the outside of the tree are not labeled. Lower level branching shows similar branching to Figure 2 but deep branching patterns (most with low support in Figure 2) shows differences, are the deep branching patterns better supported in the BUSCO tree?

L. 343 it can be difficult to draw conclusions from absence of functions in incomplete bins.

I. 375 'straightening' possibly a typo.

L. 392 references non-existent table S8. Likely meant to reference table S6. Column names in table S6 are of unclear meaning

Figure 4: Out of our expertise, but may be better suited as a supplemental figure or would benefit from more description of the approach used in the main text.

Reviewer #2 (Comments to authors)

This manuscript reports the analysis of a massive sequencing effort from marine planktonic organisms (about 1000 metagenomes over the global sunlit ocean and across multiple size fractions) with the aim to reconstruct MAGs (metagenome assembled genomes) of ecologically relevant microbial eukaryotes. This extensive effort seems successful in recovering a large set of new genomes of many species that are still uncultured and uncharacterized. The paper is relatively well written, the dataset is impressive, but the ecological analysis performed could be improved, while there is the fundamental lack of a proper evaluation of the quality and credibility of the MAGs retrieved. I suggest major revision.

My main concern on this manuscript is that there is a lack of a direct evaluation of the quality of the MAGs. Do they represent unique species, or the mix of very closely related species? It is intriguing, for instance, to find 20 *Micromonas* MAGs, when the current understanding of the diversity of this genus includes less than 10 species. This quality analysis could be done by comparing the MAGs with available

genomes of the few species that appear in both datasets. Interesting cases could be *Micromonas*, *Ostreococcus*, *Bathycoccus*, *Chloropicon*, *Cafeteria*, *MAST-4*. Some of these genomes represent widely distributed marine species, so they should be found in the MAGs list. Best results would be that the complete MAG is a subset of the standard genome.

The MAG approach, very successful for prokaryotes and viruses, seemed less promising for microbial eukaryotes, due to genome size and complexity. Here, you present a very interesting SAG collection, but it is not clear why you were so successful. It could be the massive sequencing effort (almost 1000 metaGs), improved binning procedures (sequence composition, GC content, differential coverage), the manual curation effort, or the use of reference data (taxonomic signal using genomic databases). So, is there any of these reasons that has a larger weight in your success?

I understand the concept of functional convergence and the effort to differentiate between vertical evolution and habitat adaptation in defining the grouping of marine species according to their gene functions. I can see examples of this functional convergence, and also examples of vertical evolution, so I disagree with the strong emphasis on the first, which appears as the main message of the paper (even at the title). Looking at the figures, there is a lot of evolutionary grouping, as most taxonomic clades are part of the same functional group, and also cases of non-evolutionary groupings. Also, how strong are the 4 clades identified? Do they have robust statistical support?

The name SMAGs is confusing, I initially thought that it referred to a particular case of MAGs that were better constructed because they used SAG information. Instead, SMAGs are the simple sum of MAGs and a few SAGs from the same samples. In fact, I do not see why SAGs are added in this dataset: the very relevant data from the paper are the 683 non-redundant MAGs, and I would focus on this dataset. And then, for particular analysis, you could add the SAGs if you feel convenient.

Including the animals with protists does not help in the analysis of the novelty of the new genomic resource. First, the analysis of the metagenomic representation of SMAGs and METdb is a bit unfair, as the METdb targets does not include animals. Second, when displaying the SMAGs novelty in Fig. 2, the reference genomes existing for copepods are not included. So, I would treat animals separately. I found interesting to exploit the metagenomes for animal genomics, but there are more direct ways to obtain their genomes (picking single individuals).

The model for present and future distributions of species in the global ocean is very interesting and suggestive, but should be better explained and sustained. At least, the present day distribution could be compared with the relative abundances observed. Even though environmental parameters (which I understand is what drives the model) are very important for microbe distribution, I guess there are not the unique factors.

Minor comments

Add "sunlit" in the title". Also, as mentioned before, I would not use "functional convergence" as the main message of the paper

The number of metagenomes included in the analysis varied from 943 (page 14), to 939 (Table S1), and to 798 (in page 3). Please clarify.

Line 140. I could not find these three unclassified MAGs in Table S3

Lines 250-251. I do not understand the logics of the "sister clades" to chrysophytes, Phaeocystis, or Pycnococcus. These MAGs may simply represent species from the group without representative genomes.

Line 252. There are multiple MAST clades, and some of them are close to MAST-4. Why all of them are interpreted as MAST-4?

Lines 261-265. Truly dinoflagellates is an important group missing from the MAG collection, but it is not the only one. Other relatively abundant groups not found here are MALVs, acantharea, radiolaria, other MASTs, Telonema, Picozoa, diplomids. You could analyze the Tara metabarcoding data (including multiple size fractions) to better identify what is missing in your list.

Lines 330-357. It is nice to identify gene functions for clades C and D. Could this be done for clades A and B as well?

Lines 470-473. I disagree here. Part of the functional grouping is driven by the "evolutionary history" (as commented above), and part is also driven by trophic modes (i.e., clade C seem obligate photoautotrophs).

Authors' response to the first round of review

Author responses to reviewer #1

Delmont et al. report the recovery of over 700 environmental eukaryotic genome bins from a combination of Tara Oceans shotgun metagenomic sequencing and single cell genomics samples. The binned genomes greatly expand the sequenced diversity of ocean eukaryotes, particularly of copepods. Further, these genome bins provide important references for existing Tara Oceans eukaryotic transcriptomic and gene centric approaches, allowing for analysis of the functional repertoire of organisms represented in these datasets. Functional classification and clustering of the eukaryotic bins revealed four primary functional groups, composed of distantly related lineages and suggesting functional convergence, but is a result that may be unduly influenced by completeness of the individual bins. A highlight of this report is that it brings important new approaches for the manual curation and binning of eukaryotic genomes and provides a detailed methodology, a valuable resource for the community.

We thank the reviewer for their interest in our methodology, genomic resource and scientific insights. We perfectly agree that completion of MAGs is an important aspect to consider.

As the reviewer noted, completion estimates vary greatly between MAGs. As a result, we have shared the concerns about a possible impact of genomic completion on the functional clustering outcomes. First, from the genomic functional profiles in Table S5 and clustering results illustrated in Figures 2, S4 and S4, it is clear that the high functional redundancy (functions detected multiple times in the same MAG or

SAG) of eukaryotic lineages (far more important as compared to bacterial and archaeal lineages) linked MAGs from the same lineage regardless of their completion values. This important aspect of the data was probably not sufficiently presented in our first submission of the study. We have now computed the functional redundancy and summarized it in the main text (see point #1 below for details). We also better described how MAGs and SAGs with similar taxonomy are connected despite great variations in completion estimates (see point #2 below for details). Finally, we also did the following verification: we performed a clustering based only on MAGs and SAGs of higher completion, which not only perfectly recapitulated our findings but also better demonstrate the homogeneous completeness signal in our clustering outcome (see point #3 below for details). We have added these results in the main text (see **bold sections**):

*“With EggNOG (refs 52–54), we identified orthologous groups corresponding to known ($n=15,870$) and unknown functions ($n=12,567$, orthologous groups with no assigned function at <http://eggno5.embl.de/>) for 4.7 million genes (nearly 50% of the genes, see Methods). **Among them, functional redundancy (i.e., a function detected multiple times in the same MAG or SAG) encompassed 46.6% to 96.8% of the gene repertoires (average of 75.2% of functionally redundant genes).** We then used these gene annotations to classify the MAGs and SAGs based on their functional profiles (Table S5). Our hierarchical clustering analysis using Euclidean distance and Ward linkage (an approach to organize genomes based on pangenomic traits (ref 55) first split the MAGs and SAGs into small animals (Chordata, Crustacea, copepods) and putative unicellular eukaryotes (Figure 3). Fine-grained functional clusters exhibited a highly coherent taxonomy within the unicellular eukaryotes. For instance, **MAGs affiliated to the coccolithophore *Emiliana* (completion ranging from 7.8% to 32.2%), Dictyochophaceae family (completion ranging from 8.6% to 76.9%) and the sister clade to *Phaeocystis* (completion ranging from 18.4% to 60.4%) formed distinct clusters. The sister clade to *Cryptophyta* (completion ranging from 1.6% to 75.7%)** was also confined to a single cluster that could be explained partly by a considerable radiation of genes related to dioxygenase activity (up to 644 genes). Most strikingly, the Archaeplastida MAGs not only clustered with respect to their genus-level taxonomy, but the organization of these clusters was highly coherent with their evolutionary relationships (see Figure 2), confirming not only the novelty of the sister clade to *Pycnococcus*, but also the sensitivity of our framework to draw the functional landscape of unicellular marine eukaryotes. **Clearly, the important functional redundancy of MAGs and SAGs minimized the effect of genomic incompleteness in our efforts assessing the functional profile of unicellular marine eukaryotes.**”*

*“Four major functional groups of unicellular eukaryotes emerged from the hierarchical clustering (Figure 3), which was **perfectly recapitulated when** incorporating the standard culture genomes matching to a MAG (Figure S9), and when **clustering only the MAGs***

and SAGs >25% complete (Figure S10)."

In conclusion, we found that within the scope of our database (MAGs and SAGs >10 Mbp in length) the genomic incompleteness does not prevent sensitive clustering of unicellular eukaryotic populations based on the functional profile of their associated MAGs. We thank the reviewer for having suggested those additional analyses, which are clarifying the important aspect of functional redundancy in marine unicellular eukaryotic genomes.

Point #1: A considerable functional redundancy in MAGs and SAGs: We found that an average of >75% of genes in each MAG and SAG were linked to a functional annotation occurring multiple times in that environmental genome. This important functional redundancy in eukaryotic genomes is minimizing the effect of MAG incompleteness and helped linking MAGs corresponding to the same lineage regardless of their completion values (see point #2). The table S5 provides access to the functional occurrence data revealing the redundancy of many functions. We have also incorporated this important aspect of the data in the main text.

Point #2: MAGs are organized based on taxonomy rather than completion within each functional group: We have displayed MAG completion in Figure 3 ("classical" clustering based on EggNog functions), and the supplemental figures S3 (clustering based on gene clusters of known and unknown functions) and S4 (clustering based only on the gene clusters of unknown function). For all those clustering outcomes, functional groups (A, B, C and D) contain both high completion and low completion MAGs (this information is available in table S3). Critically, MAGs and SAGs are organized based on taxonomy rather than completion within each functional group. Some striking examples include the sister clade to *Phaeocystis* (completion ranging from 18.4% to 60.4%), Cryptophyta (completion ranging from 11.8% to 69.8%), Dictyochophaceae family (completion ranging from 8.6% to 76.9%), and the sister clade to Cryptophyta (completion ranging from 1.6% to 75.7%). Despite considerable differences in completion estimates, these lineages are organized in distinct, clearly defined clusters in our analyses (Figures 2, S3 and S4). This is now described in the text, and explained by the important functional redundancy.

Point #3: Functional clustering using only high completion MAGs and SAGs supports our conclusions:

Functional clustering based on 483 MAGs and SAGs >25% complete (230 MAGs and SAGs were removed) perfectly recapitulated the four functional groups we initially identified using the entire dataset (Figure S10).

Critically, completion estimates displayed in the above figure clearly show that genomic incompleteness does not impact the formation of the four functional groups described in our study, confirming what was also described regarding Figure 2. Again, this can easily be explained by the important functional redundancy of MAGs and SAGs, now introduced in the main text.

At its core, this paper addresses the functional and phylogenetic diversity of microbial eukaryotes in the oceans. The methods are modified from standard bacterial shotgun metagenomic methods to be applied to microbial eukaryotes in a

novel way. The authors report important findings, however there are areas in need of improvement. In addition, the manuscript would benefit from more clear or expanded justification on why some choices were made, such as the inclusion of low completeness bins which are typically filtered, the choice of proteins for Figure 2, and the initial decision to use a co-assembly based approach.

We have better explained the methodology in the main text and supplemental information document, and further address the comments in the three following sections:

Inclusion of low completeness bins:

Given the complexity of eukaryotic genomes and current scarcity of reference marine genomes, it is not uncommon to perform analyses on low completion environmental genomes (e.g., with single cell sorting) to shed some lights into their functioning. As described above, we now have demonstrated that the least complete MAGs and SAGs did not negatively impact the functional clustering trends, instead providing more diversity to best address the genomic functional landscape of unicellular eukaryotes in the sunlit ocean. This is now better explained in the text.

The choice of proteins for our main phylogenetic analysis:

Regarding choice of proteins for phylogenetic analyses, we used in our prime analysis the six manually curated RNA polymerase genes (types I, II and III of the subunits A and B). Those are extensively studied gene markers (e.g., see <https://doi.org/10.3389/fmolb.2021.663209> and <https://www.nature.com/articles/nrmicro2507> for some perspectives). The corresponding proteins are often used as references for phylogenetic and phylogenomic analyses given their lengths (multi kb genes) and evolutionary stability over long periods of time. For example, a recent study dedicated to comparing marker genes for multi-domain phylogenetic reconstructions concluded that the RNA polymerase genes were the best markers within the scope of their study (<https://doi.org/10.1093/molbev/msab254>), likely due to lower rates of lateral gene transfers (lower incongruences). RNA polymerase genes are often overlooked when performing eukaryotic phylogenomic analyses because they are at times fragmented, and because three types are present in eukaryotic genomes (types I, II, and III), each containing two subunits. Here, we inspected those proteins for each MAG and SAG as well as transcriptomes from METdb, manually fixed fragmentation, identified the types I, II and III, and removed duplicates. This allowed us to describe the phylogenetic relationship of eukaryotes from MAGs, SAGs and cultures with high resolution. This is now better explained in the text:

*“METdb was chosen as a taxonomically curated reference transcriptomic database from culture collections, and the two largest subunits of the three DNA-dependent RNA polymerases (six multikilobase genes found in all modern eukaryotes and hence already present in the Last Eukaryotic Common Ancestor). **These genes are highly relevant markers for the phylogenetic inference of distantly related microbial organisms (ref 42) and contributed to our understanding of eukaryogenesis (ref 43). They have long been overlooked to study the eukaryotic tree of life, possibly because automatic methods are currently missing to effectively identify each DNA-dependent RNA polymerase type prior to performing the***

phylogenomic analyses. Here, protein sequences were identified using HMMs dedicated to the two largest subunits for the MAGs and SAGs (n=2,150), and METdb reference transcriptomes (n=2,032). These proteins were manually curated and linked to the corresponding DNA-dependent RNA polymerase types for each subunit using reference proteins and phylogenetic inferences (see Methods and Supplemental Material)."

Note that we also used the BUSCO single copy core genes as a complementary analysis, however those genes were not manually curated due to the extended number of genes involved. The BUSCO-centric phylogenomic analysis is used as a complementary approach that supported most of the trends that emerged from the RNA polymerase genes. BUSCO approach included less MAGs and SAGs due to a lack of marker genes identified. Furthermore, not all BUSCO gene markers are as good as the RNA polymerase genes for phylogenetic analyses (see <https://doi.org/10.1093/molbev/msab254> as a recent study example on this front). We thank the reviewer for helping clarifying this important aspect of our study: the manual curation and identification of thousands of proteins corresponding to RNA polymerase genes and found in environmental genomes and reference transcriptomes.

The metagenomic co-assemblies:

There is a long history of debates regarding the advantages and limitations of both single assemblies and co-assemblies. As the reviewer noted later on, co-assemblies increase the complexity of the data (more microdiversity traits and more populations covered), challenging the assembly step. On the other hand, co-assemblies can provide more coverage of a given genome, allowing recovery of MAGs that are detected in multiple samples but never with sufficient coverage for a recovery with the single assembly strategy. In addition, co-assemblies have the critical advantage of minimizing number of assembly outputs, and hence the overall binning and curation efforts. With a dataset of nearly 1,000 metagenomes, we only binned 11 co-assemblies, allowing a careful manual binning and curation that would otherwise not been possible within the temporal scope of our project. Indeed, it required ~1 year to manually bin and curate those large co-assemblies. In the past, we have reconstructed a large number of bacterial MAGs using this co-assembly strategy, on both the small size fraction (<https://www.nature.com/articles/s41564-018-0176-9>) and more recently the larger size fractions (<https://doi.org/10.1038/s41396-021-01135-1>). In the case of eukaryotes, genomic coverage is even more critical since the genomes are often very large. In fact, about half of the MAGs would not have been recovered using a single assembly approach, provided that a minimum of 10x coverage is required during the assembly step. This section of our results emphasizes this important point favouring the use of co-assemblies: *"Nearly half the MAGs did not have vertical coverage >10x in any of the metagenomes, emphasizing the relevance of co-assemblies to gain sufficient coverage for relatively large eukaryotic genomes."*

To further clarify our strategy and acknowledge possible limitations, we have also added this in the main text (**in bold**):

"We used the 280 billion reads as inputs for 11 metagenomic co-assemblies (6-38 billion reads per co-assembly) using geographically bounded samples (Figure 1, Table S2), as previously done for the Tara

Oceans 0.2–3 μm size fraction enriched in bacterial cells (27). We favored co-assemblies to gain in coverage and optimize the recovery of large marine eukaryotic genomes. However, it is likely that other assembly strategies (e.g., from single samples) will provide access to genomic data our complex metagenomic coassemblies failed to resolve. In addition, we used 158 eukaryotic single cells sorted by flow cytometry from seven Tara Oceans stations (Table S2) as input to perform complementary genomic assemblies (see Methods)."

We agree that testing many assembly strategies would certainly be of interest. Yet, this endeavour would require many months of analysis and goes beyond the scope of our study, which focuses on scientific insights rather than methodology improvements, and provides a first example of what genome-resolved metagenomics of eukaryotic can provide when applied at large scale.

Major points of criticism that should be addressed:

The bin completeness distribution should be clarified in the main text. Upon first reading, it was not clear the 700+ genome bins range widely in completeness (0% to 93.7% BUSCO completeness, Table S3), information that is needed to properly interpret the main findings of the manuscript. The authors use a minimum bin size cutoff (10 mbp, l. 118) for filtering eukaryotic bins of interest, but this alone is not sufficient to make the quality of the individual genomes interpretable as eukaryotic genome size is independent of gene content and thus, genome completeness.

We totally agree that a minimal MAG length during binning does not guaranty a minimal completion score after genes are properly identified. This is indeed confirmed for our database. However, we see value in all the >10 Mbp MAGs and SAGs combined in our database. For example, most of them could be included in our phylogenetic analysis based on the curated RNA polymerase genes. In the initial version of our manuscript, the completion values were available in Supplementary Table S03. We have now clarified the bin completeness variations directly in the main text (**in bold**):

*"This new genomic database for eukaryotic plankton has a total size of 25.2 Gbp and contains 10,207,450 genes according to a workflow combining metatranscriptomics, ab-initio, and protein-similarity approaches (see Methods). **Estimated completion of the Tara Oceans MAGs and SAGs averaged to ~40% (redundancy of 0.5%) and ranged from 0% (a 15 Mbp long Opisthokonta MAG) to 93.7% (a 47.8 Mbp long Ascomycetes MAG).** Genomic lengths averaged to 35.4 Mbp (up to 1.32 Gbp for the first Giga-scale eukaryotic MAG), with a GCcontent ranging from 18.7% to 72.4% (Table S3)."*

This was indeed an important clarification to make.

L. 294 Given the wide range of genome completeness across included SMAGs, genome completeness is likely to influence clustering based on presence or absence of functional repertoire. As written, it is not clear this is controlled for in the current analysis. Do the same patterns of clustering (general structure of the four major groups) hold when excluding low completeness bins? Do the functional groups

retain their structure when closely related cultured representatives are included?

As described in more details in our previous responses to the reviewer, our functional clustering based on MAGs and SAGs with >25% completion perfectly recapitulated the emergence of the four functional groups (see figure S10), and clearly shows that clustering is not governed by completion.

Regarding the integration of closely related cultured representatives, we had already described in our initial manuscript that we found a total of 24 matches between MAGs and METdb transcriptomes from culture (ANI >98%, see Table S3). Since not all transcriptomes have a corresponding genome, we could only retrieve culture genomes for *Bathycoccus prasinos*, *Micromonas commoda*, *Micromonas pusilla*, *Ostreococcus lucimarinus*, *Pycnococcus provasolii* and *Cafeteria roenbergensis*. By aligning the MAG and corresponding culture genome for each pair, we could quantify the Average Nucleotide Identity (ANI) and portion of alignments, now summarized in Table S4 and Figure S1.

Critically, MAGs corresponded to a subset of the “standard” culture genomes for each match (Table S4). The portion of MAGs that could be aligned to the culture genomes ranged from nearly 100% (e.g., for *Ostreococcus lucimarinus*) to less than 40% (the case of *Cafeteria roenbergensis*). These matches provide striking examples of how similar MAGs and culture representatives can be.

We have also computed the functional profile of the culture genomes matching a MAG, and incorporated them into our functional profile. As expected, the genomes from cultures were organized next to their corresponding MAGs, and did not impact the overall structure of the clustering (Figure S9).

L. 108 Large co-assemblies can assemble poorly due to introduced heterogeneity, and may explain some of the highly incomplete genomes. Has targeted reassembly of individual samples or smaller co-assemblies been attempted? If not, assembling and binning a subset of samples would be useful for comparison, particularly because this paper will serve as an important reference for methodological approaches and knowing the potential trade-offs and benefits of co-assembly compared to deep sequencing of individual samples will be beneficial.

Our decision to favour co-assemblies is now better explained in the main text (see previous responses to the reviewer for more details). In addition, we did perform one targeted genome-resolved metagenomic effort to improve completion of the largest MAG ever recovered (Southern Ocean region). Instead of co-assembling all 19 metagenomes for this region, we only co-assembled metagenomes for which this MAG had a coverage superior of “1x” and considered smaller contigs as well. We gained a substantial portion of the MAG using this approach, however high fragmentation remained. Here is the method section describing this work:

“A first Giga scale eukaryotic MAG. We performed targeted genomeresolved metagenomics to confirm the biological relevance and improve statistics of the single MAG longer than 1 Gbp with an additional coassembly

(five Southern Ocean metagenomes for which this MAG had an average vertical coverage >1x) and by considering contigs longer than 1,000 nucleotides, leading to a gain of 181,8 million nucleotides. To our knowledge, we describe here the first successful characterization of a Gigabase-scale MAG (1.32 Gbp with 419,520 scaffolds), which we could identify using two distinct metagenomic co-assemblies.”

We understand the reviewer’s motivation for a more in depth comparison of methods to recover eukaryotic MAGs. Unfortunately, while it is theoretically possible to gain completion of hundreds of MAGs with targeted genome-resolved metagenomics, our manual binning and curation protocol would take years to complete. A methodological study describing in details pros and cons of single assemblies versus co-assemblies for eukaryotic genomics is certainly of interest, however this would be best assessed using automatic binning. This falls beyond the scope of our study, which focuses on the evolution and ecology of marine eukaryotes rather than methodology. We hope that the reviewer and the editor will appreciate our reasoning, and see value in both our database and scientific insights.

Minor points:

L. 137 It is interesting there were bins with such low GC content given illumina sequencing bias. Are these bins typically low completeness or low coverage? It may be worth noting them separately

Illumina sequencing is known to introduce GC-content biases. However, past studies showed we could recover low and high GC-content marine MAGs. For example, among the 1,888 bacterial and archaeal MAGs we have recovered from the sunlit ocean, GC-content ranged from 24% to 74% (<https://www.biorxiv.org/content/10.1101/2021.03.24.436778v1.abstract>). Here, GC-content of eukaryotic MAGs and SAGs ranged from 19% to 72%, in line with the bacterial and archaeal MAGs. In addition, we did not see any obvious link between GC-content and coverage or completion (Table S3), as summarized here.

L. 166 Metagenomics has the potential to better represent environmental eukaryotic diversity than culturing, but mapping reads from the samples you assembled your genomes from and comparing to cultured genomes is not an unbiased way to make this argument. Are there representative ocean samples from which you didn't assemble that could be used for mapping?

This is a very good point. Needed are deeply sequenced metagenomes enriched in eukaryotic signal for the sunlit ocean (large size fractions). We could not find such datasets outside the scope of the Tara consortium, and as a result we focused on unpublished data from *Tara* Pacific. Briefly, we performed an additional mapping (same methodology as in our manuscript) onto 62 metagenomes (from 148 to 667 million Illumina reads per sample) corresponding to large cellular size fractions (0.2 – 3 μm , 3 – 20 μm , > 20 μm and > 300 μm) of 18 stations collected by the *Tara* consortium. This sampling was done during the first leg of the *TARA* Pacific project when crossing the Atlantic Ocean (from France to Panama), years after *Tara* Oceans. These Atlantic stations were not used to characterize the MAGs. The corresponding metagenomes are currently not publically available. While we cannot at this time

release those datasets, the results support our statement: MAGs and SAGs better represent plankton in the sunlit ocean as compared to METdb.

General statistics on cultured ocean eukaryotic genome structure, such as genome size, GC, etc, would be useful for context and comparison to the SMAGs.

Assuming the reviewer refers to the METdb database for cultures, presented in the Table S3, those are transcriptomes, so we cannot display genomic length, GCcontent or general gene calling statistics. Simply, the two types of data are too different for a meaningful comparison of such genomic metrics. On the other hand, we worked on a common taxonomical framework for both databases, available in the Table S3.

L. 261 How large of a role do the authors think extraction played in the types of genomes recovered? Could this partially explain the missing Dinoflagellata?

While DNA extraction is never perfect, the one elected by the *Tara* Oceans consortium is commonly used for marine microbial life. Based on the same extracted DNA, 18S rRNA gene amplicon surveys detected a considerable amount of Dinoflagellata (see <https://www.science.org/doi/10.1126/science.1261605>), so the lack of MAGs for this particular lineage is not due to DNA extraction. As stated in the text, it is likely due to their considerable genomic lengths. More sequencing efforts might help recovering this important lineage. However, other factors might drive the lack of MAGs for this lineage (e.g., lack of highly abundant populations for the Dinoflagellata would make it challenging to recover MAGs).

Figure 2: It is difficult to spot the putative new group in the tree.

We created a supplemental figure (Figure S8) to better visualize the putative new group and included it in the main text

Finally, we also better emphasize this lineage in the BUSCO phylogeny (see Figure S2, also described in the next section).

Supplemental figure S1: branch supports and additional tracks on the outside of the tree are not labeled. Lower level branching shows similar branching to Figure 2 but deep branching patterns (most with low support in Figure 2) shows differences, are the deep branching patterns better supported in the BUSCO tree?

We thank the reviewer for allowing us to significantly improve the Figure S2 (figure S1 in the initial submission). We have added the support values and labels.

Many deep branching patterns are better supported. However the putative new group branching is not well supported, similar to what we observed with the RNA polymerase genes. This is intriguing, and we are confident more genomic data and other analyses will help solve this in future studies.

L. 343 It can be difficult to draw conclusions from absence of functions in incomplete bins.

We agree that when looking at individual MAGs, the absence of a function does not mean it is indeed absent from the corresponding population, due to incompleteness.

On the other hand, clear trends can emerge from the analysis of hundreds of MAGs. In our case, MAGs were generally of good completion within Group C (average completion of 63%) and many MAGs were of lower completion in the other groups (A, B and D). Yet we saw a scarcity of some interesting functions in Group C, compared to the other groups (supported by statistics). Clearly, these insights could not be explained by genomic incompleteness. As a result, we are confident those trends and other summarized in Table S6 are biologically relevant.

I. 375 'straightening' possibly a typo.

We replaced it by “reinforcing”.

L. 392 references non-existent table S8. Likely meant to reference table S6. Column names in table S6 are of unclear meaning

We had a Table S8 (now labelled S9) with the following legend: “Niche partitioning and world map projection statistics for 374 MAGs and SAGs”. It is unfortunate that the reviewers might not have been able to access this information. Table S6 (now Table S7) relates to the AGNOSTOS gene clusters for the ~10 million genes found in our database.

Figure 4: Out of our expertise, but may be better suited as a supplemental figure or would benefit from more description of the approach used in the main text.

A more detailed description is now available in the supplemental information document.

Author responses to reviewer #2

This manuscript reports the analysis of a massive sequencing effort from marine planktonic organisms (about 1000 metagenomes over the global sunlit ocean and across multiple size fractions) with the aim to reconstruct MAGs (metagenome assembled genomes) of ecologically relevant microbial eukaryotes. This extensive effort seems successful in recovering a large set of new genomes of many species that are still uncultured and uncharacterized. The paper is relatively well written, the dataset is impressive, but the ecological analysis performed could be improved, while there is the fundamental lack of a proper evaluation of the quality and credibility of the MAGs retrieved. I suggest major revision.

We thank the reviewer for seeing value in our methodology and database. We agree with the reviewer that MAG and SAG quality evaluation is critical. In our study, we have dedicated a significant amount of time in manual binning and curation (state-of-the-art approach for quality), inspecting all the MAGs and SAGs one by one using an advanced visualization strategy.

We have addressed the evaluation and credibility critic of our database in the three following sections. They cover both environmental and genomic signal we used to gain confidence in the quality of the MAGs and SAGs we have characterized.

Curation and quality assessment of the MAGs based on mapping results:

Curation and quality assessment of the MAGs based on mapping results was described in detail in the supplementary material document:

“Within the framework of our study, the anvio interactive interface took

*advantage of the sequence composition of contigs, their differential coverage across metagenomes, taxonomic signal using a reference database that includes METdb, and HMM models for single copy core gene collections (Bacteria, Archaea, Eukarya). When selecting a cluster of contigs corresponding to a MAG in the interface, anvio identified its domain affiliation in real time using random forest, and displayed its completion and redundancy values accordingly. This way, it was possible to focus on the eukaryotic MAGs within an assembly containing also many abundant bacterial and archaeal MAGs. In the figure 3, we provide the example of one CONCOCT cluster from the Mediterranean Sea metagenomic co-assembly (95 metagenomes) containing eukaryotic MAGs for *Ostreococcus* and *Micromonas* (left panel). In this simple example, we selected those two clusters in the interface, saved the collection, and subsequently manually curated them as presented here for *Ostreococcus* (right panel). This MAG exhibited a completion of 100% and a redundancy of 3%. One metagenome (most outer blue layer) was particularly useful in this particular case since the *Micromonas* MAG was more detected compared to the *Ostreococcus* MAG, allowing an effective binning outcome. Given the complexity of marine metagenomes, differential coverage across dozens of metagenomes strongly benefited to the outcome of our genome-resolved metagenomic survey.”*

We have now expanded this section with the example of another manually curated MAG, for which environmental signal is described using both detection (horizontal coverage, left panel) and mean coverage (vertical coverage, right panel).

The coherence of environmental signal is supportive of the quality of this MAG and others. Videos have been made available to explain in more details how the interactive interface can be used to spot potential MAG contaminations and perform curation : <https://merenlab.org/2017/01/03/loki-the-link-archaea-eukaryota/>

Curation and quality assessment of the SAGs based on mapping results:

Curation and quality assessment of the SAGs based on mapping results was also described in detail in the supplementary material document:

“Eukaryotic single cell genomes (SAGs) can be heavily contaminated due to a combination of factors during cell sorting, DNA extraction and amplification, and multiplex sequencing. Here, we slightly modified the anvio metagenomic workflow to effectively decontaminate marine eukaryotic SAGs, one by one. Briefly, we used the anvio interactive interface to manually curate eukaryotic SAGs by taking into consideration the sequence composition of contigs, their differential coverage across 100 most relevant metagenomes (i.e., those with highest mapping recruitment scores within the scope of TARA Oceans), taxonomic signal using a reference database that includes METdb, and HMM models for single copy core gene collections (Bacteria, Archaea, Eukarya). Note that compared to the metagenomic co-assemblies, the number of contigs under consideration was orders of magnitude

smaller. Since all contigs could be loaded in the interactive interface, there was no need to use the pre-clustering step with CONCOCT. However, CONCOCT could also be used here if some SAG assemblies include more than ~25k contigs.

Figure 6 provides a striking example of heavily contaminated SAG we could effectively curate thanks to the clear differential coverage signal of contigs across 100 metagenomes. In this particular case, contamination seemed to have multiple origins, and a large number of contigs were removed. Overall, our manual curation of SAGs using a genome-resolved metagenomics workflow initially built for MAGs turned out to be highly valuable, leading in our study to the removal of more than one hundred thousand scaffolds for a total volume of 193.1 million nucleotides. This metagenomic-guided decontamination effort contributes to previous efforts characterizing eukaryotic SAGs from the same cell sorting material (refs 8–12) and provides new guidelines for marine eukaryotic SAGs. We now recommend this approach for future efforts generating eukaryotic SAGs from the sunlit ocean. This is important, especially since SAGs could become a valuable asset in the near future to target lineages genome-resolved metagenomics failed to recover so far. It is especially the case of Dinoflagellates.”

Thanks to our workflow, we realized that marine eukaryotic SAGs could be heavily contaminated and that attempts to automatically remove contaminants were lacking the required resolution. Here, we solved this problem by applying genomeresolved metagenomic tools to the SAGs abundant in the sunlit ocean. In our view, this is a critical methodological improvement for quality purposes.

Quality assessment of the MAGs and SAGs using genomic statistics:

In addition to the environmental signal, we used the completion and redundancy estimates based on both BUSCO (automatically identified single copy core genes) and the RNA polymerase genes (six markers that we manually curated for our phylogenetic analysis) to assess the quality of the MAGs and SAGs (summary in Table S3). Strikingly, there was an average of just 0.4% of redundant BUSCO single copy core genes in the MAGs and SAGs. Those results point to a minimal amount of possible contaminations in the database.

Overall, based on environmental signal, genomic statistics, and comparisons with genomes from cultures (see the following section), we are highly confident in the quality of our curated MAGs and SAGs, and consider we have extensively described this quality aspect of the database. We hope the reviewer and the editor also see value in our careful manual binning and genomic curation efforts, and the extensive supporting evidence compiled in our supplementary documents and tables.

My main concern on this manuscript is that there is a lack of a direct evaluation of the quality of the MAGs. Do they represent unique species, or the mix of very closely related species? It is intriguing, for instance, to find 20 *Micromonas* MAGs, when the current understanding of the diversity of this genus includes less than 10 species. This quality analysis could be done by comparing the MAGs with available genomes of the few species that appear in both datasets. Interesting cases could be *Micromonas*, *Ostreococcus*, *Bathycoccus*, *Chloropicon*, *Cafeteria*, *MAST-4*. Some of these genomes represent widely distributed marine species, so they should

be found in the MAGs list. Best results would be that the complete MAG is a subset of the standard genome.

The reviewer rightfully questions the biological meaning of MAGs, which emerge from the assembly of billions of metagenomic reads and aim at representing a consensus genomic sequence of closely related cells corresponding to a same “population” abundant in parts of the sunlit ocean. MAGs have considerably expanded the known diversity of bacteria, archaea, and viruses in the past decades and it is not surprising that our study shows similar trends for the eukaryotes. We have addressed the biological relevance of the MAGs in the two following sections. First section covers genomic comparisons indicating that MAGs with matches to cultivation are a subset of “standard” culture genomes. Second section covers biogeography investigations confirming that we expanded the genomic landscape of various genera (including on the front of critical, well-known photosynthetic genera) as compared to cultivation.

Complete MAGs are a subset of standard genomes from culture

The reviewer asks if “*the complete MAG is a subset of the standard genome*”. As already described in our initial manuscript, we found a total of 24 nearly identical matches between MAGs and METdb transcriptomes from culture (ANI >98%, see Table S3). Since not all transcriptomes have a corresponding genome, we could only retrieve culture genomes for *Bathycoccus prasinos*, *Micromonas commoda*, *Micromonas pusilla*, *Ostreococcus lucimarinus*, *Pycnococcus provasolii* and *Cafeteria roenbergensis*. By aligning the MAG and corresponding culture genome for each pair, we could quantify the Average Nucleotide Identity (ANI) and portion of alignments, summarized in Table S4 and Figure S1.

Critically, for each match considered the MAG corresponded to a subset of the “standard” culture genome (Table S4). The culture genomic portion covered by the MAG varied from nearly 100% (e.g., for *Ostreococcus lucimarinus*) to about one third (36.5 % in the case of *Cafeteria roenbergensis*). These matches provide striking examples of how similar eukaryotic MAGs and culture representatives can be, coherent with what was previously observed with marine bacterial MAGs and reference genomes (e.g., for the diazotroph UCYN-A, see <https://www.nature.com/articles/s41564-018-0176-9>).

Regarding the example of *Pycnococcus*, we described two MAGs that are closely related to the reference genome (Figure S1). However, the MAG from match #2 is slightly more distant (ANI score <98%), and has a much larger portion of its genomic length not matching to the reference (>30%), suggesting it corresponds to a distinct population for which we do not have access to a reference culture genome. When comparing the distribution of the two MAGs across *Tara* Oceans metagenomes, the R2 is only of 0.67, confirming they correspond to two distinct populations of *Pycnococcus*.

We thank the reviewer for addressing the similarity between MAGs and standard culture genomes. Those results contribute to demonstrating the biological relevance of our genomic database for marine unicellular eukaryotes.

Most MAGs correspond to distinct eukaryotic populations

The reviewer asks if MAGs “*represent unique species, or the mix of very closely related*

species". As an important reminder, we have removed redundancy so that all MAGs and SAGs in our study have an average nucleotide identity (ANI) <98%.

Defining the boundaries of species is an exciting research venue that is not trivial and goes well beyond the scope of our study. While some experts favour an ANI cutoff of 95% instead of 98%, microbial MAGs with ANI between 95% and 98% can display distinct distribution patterns in the sunlit ocean. We used the 98% ANI cutoff to prevent merging MAGs with distinct distribution patterns, since they might probably represent distinct populations. As a result, some of the most closely related populations may fall within the scope of the same "species".

To explore this further and determine whether MAGs in our database "*represent unique species, or the mix of very closely related species*", we have analysed the genomic distances and biogeography of environmental genomes corresponding to *Micromonas* (20 non-redundant MAGs), *Bathycoccus* (8 non-redundant MAGs), *Ostreococcus* (4 non-redundant MAGs), *Chloropicon* (11 non-redundant MAGs) and MAST closely related to MAST-4 and MAST-7 (26 non-redundant MAGs and SAGs), as suggested by the reviewer (Cafeteria was excluded because we only have one MAG for this genus, however its linkage to the culture genome is displayed in Figure S1 described in the above section). If two environmental genomes share a very similar distribution pattern and genomic content, then it is likely they correspond to the same "population", or "species". As cut-offs, we used a coefficient of determination (R^2) > 0.9 for the mean coverage values across metagenomes (very similar distribution patterns), and ANI >95% (often used to delineate microbial species).

In the case of *Micromonas*, we found that at least 16 populations occur with distinct distribution patterns in the sunlit ocean (Figure S4, top panel), indicating that our database goes well beyond the known diversity of *Micromonas*. Just four of the 20 MAGs were matching to a known *Micromonas* culture (ANI >98% with METdb transcriptomes). Interestingly, two MAGs matching to distinct METdb transcriptomes for *Micromonas commoda* shared 96.8% ANI and displayed distinct distribution patterns, with the MAG characterized from the Mediterranean Sea more detected in this regions (Figure S4, bottom panel). Importantly, other MAGs displayed various distribution patterns (e.g., in the Indian Ocean) yet lack culture representatives.

Our results support the biological relevance of the *Micromonas* MAGs, suggesting they correspond to a wide range of distinct populations, rather than "a mix of closely related species". Following the same strategy, we identified at least 5 distinct populations of *Bathycoccus* (based on 8 MAGs, see figure S6), 4 distinct populations of *Ostreococcus* (based on 4 MAGs), 11 distinct populations of *Chloropicon* (based on 11 MAGs, see figure S5), and 24 distinct populations of MAST (based on 26 MAGs and SAGs, see figure below).

Thus, the large majority of MAGs in our database appear to correspond to distinct populations that display unique distribution patterns in the sunlit ocean. Our cut-off of 98% ANI to remove redundancy appears effective, and allowed us to keep closely related MAGs with distinct distribution patterns, as exemplified by the two *Micromonas commoda* MAGs.

The MAG approach, very successful for prokaryotes and viruses, seemed less promising for microbial eukaryotes, due to genome size and complexity. Here, you present a very interesting SAG collection, but it is not clear why you were so successful. It could be the massive sequencing effort (almost 1000 metaGs), improved binning procedures (sequence composition, GC content, differential coverage), the manual curation effort, or the use of reference data (taxonomic signal using genomic databases). So, is there any of these reasons that has a larger weight in your success?

Assuming the reviewer addresses the MAGs (“SAG” was used), we believe the most critical reason for our success was the sequencing depth of *Tara* Oceans, the second being our computing capability. Binning procedures have only slightly been improved (especially, we added a collection of single copy core genes for the eukaryotes), and the taxonomic signal is only used as supplemental information for binning. The limiting step for binning is most of the time assembly outcome. Here, we completed considerable assembly efforts by combining metagenomes from the same region, thanks to extensive computing memory capabilities. The largest coassembly we successfully completed included nearly 38 billion metagenomic reads and produced 12,858,349 contigs >1kbp. Details are available in the Table S2. In our view, it all comes down to the assembly step, which relies heavily on the sequencing effort (considerable with *Tara* Oceans). The rest is not critical, and various binning strategies would work, even though we strongly favour our manual binning and curation over automatic binning approaches.

As a perspective, in years to come we expect many more projects to provide large amounts of marine eukaryotic MAG material, powered by decreasing sequencing costs and the emergence of new sequencing technologies to expend the assembly outcomes. SAGs will of course also play an important role in charting the genomic content of eukaryotic plankton. The two approaches are very complementary, in our humble opinion.

I understand the concept of functional convergence and the effort to differentiate between vertical evolution and habitat adaptation in defining the grouping of marine species according to their gene functions. I can see examples of this functional convergence, and also examples of vertical evolution, so I disagree with the strong emphasis on the first, which appears as the main message of the paper (even at the title). Looking at the figures, there is a lot of evolutionary grouping, as most taxonomic clades are part of the same functional group, and also cases of nonevolutionary groupings. Also, how strong are the 4 clades identified? Do they have robust statistical support?

We agree with the reviewer that at a finer scale clustering is mainly explained by taxonomy. In fact in the main text we take the example of green algae (Archaeplastida):

“Most strikingly, the Archaeplastida MAGs and SAGs not only clustered with respect to their genus-level taxonomy, but the organization of these clusters was highly coherent with their evolutionary relationships (see Figure 2), confirming not only the novelty of the putative sister clade to Pycnococcus, but also the sensitivity of our framework to draw the functional landscape of unicellular marine eukaryotes.”

However, the overall shape of the clustering does indicate a functional convergence

of distantly related lineages. The organisation of Stramenopiles for instance is very interesting since it occurs in groups B, C and D depending on the lineage (e.g., diatoms are in group C while Oomycota is in group D). This is an important insight from our survey. Not only are the main trends largely recapitulated when considering only genes of unknown functions (this is an exciting observation regarding the unknowns within eukaryotic genomes), but we also have thousands of functional annotations with significant distribution differences between the four groups. These functional annotations cover ~40% of the annotated genes, which were considered in this analysis, as described in the text:

“A total of 2,588 known and 680 unknown functions covering 1.94 million genes (~40% of the annotated genes) were significantly differentially occurring between the four functional groups (Welch’s ANOVA tests, p -value $<1.e-05$, Table S5). We displayed the occurrence of the 100 functions with lowest p -values in the hierarchical clustering presented in Figure 3 to illustrate and help convey the strong signal between groups.”

Thus, about 40% of the data is statistically supporting the four groups. They correspond to a wide range of functions.

Finally, our functional clustering of MAGs and SAGs >25% complete perfectly recapitulated the four groups we identified using the entire dataset:

*“Four major functional groups of unicellular eukaryotes emerged from the hierarchical clustering (Figure 3), which was **perfectly recapitulated when incorporating the standard culture genomes matching to a MAG (Figure S8), and when clustering only the MAGs and SAGs >25% complete (Figure S9).**”*

Thus, the four main functional groups and our title are well supported by multiple lines of evidence.

The name SMAGs is a confusing, I initially thought that it referred to a particular case of MAGs that were better constructed because they used SAG information. Instead, SMAGs are the simple sum of MAGs and a few SAGs from the same samples. In fact, I do not see why SAGs are added in this dataset: the very relevant data from the paper are the 683 non-redundant MAGs, and I would focus on this dataset. And then, for particular analysis, you could add the SAGs if you feel convenient.

This was indeed confusing and we have change the naming strategy. We now describe MAGs and SAGs separately throughout the text. We thank the reviewer for improving the clarity of our manuscript.

Including the animals with protists does not help in the analysis of the novelty of the new genomic resource. First, the analysis of the metagenomic representation of SMAGs and METdb is a bit unfair, as the METdb targets does not include animals. Second, when displaying the SMAGs novelty in Fig. 2, the reference genomes existing for copepods are not included. So, I would treat animals separately. I found interesting to exploit the metagenomes for animal genomics, but there are more direct ways to obtain their genomes (picking single individuals).

We agree that picking single individuals can be used to perform animal genomics. While this was not the prime target of our study, we identified two main groups of copepods (names clades A and B) based on environmental genomics that might be

of interest to various researchers. Since our initial phylogeny was only using METdb as a reference, we agree with the reviewer that our approach was not optimal to link our copepod MAGs to known species. To address this point, we have performed an additional phylogenetic analysis focused on Opisthokonta and adding the four reference copepod genomes from NCBI for which RNA polymerase genes could be identified (Figure S3). This analysis emphasizes the novelty of clade A, and links clade B to three known species.

We thank the reviewer for improving our study with relevant additional analyses.

The model for present and future distributions of species in the global ocean is very interesting and suggestive, but should be better explained and sustained. At least, the present day distribution could be compared with the relative abundances observed. Even though environmental parameters (which I understand is what drives the model) are very important for microbe distribution, I guess there are not the unique factors.

We thank the reviewer for this remark. We added a figure (see below) in the supplementary material describing the performances of the statistical models on biogeochemical model projections at locations of the training set (i.e. the Tara Oceans stations). Our models are only presence/absence models so they project probabilities of presence (not relative abundances) of a given MAG at each gridded point of the ocean based on environmental parameters. The figure 8 presents the specificity in function of the sensitivity for each model (i.e. each point is a MAG) calculated on the set of Tara stations for biogeochemical projections and for two threshold of presence detection ($p > 0.5$ and $p > 0.3$). The specificity captures the ability of the model to correctly detect absences while the sensitivity captures its capability to detect presences. Details on model computation and validation are in the supplementary material.

Globally, models performed well, especially for $p > 0.3$ as a presence threshold, with a vast majority of models with sensitivity > 0.6 and sensibility > 0.6 (61% for $p > 0.3$, 39% for $p > 0.5$). Lowering the presence threshold allows a global increase in sensitivity with a relatively low decrease in specificity (red points versus green points). Some models perform relatively poorly and have low sensitivity. This might be explained by the asymmetry in number of presences compared to absences in the training set (relatively many more absences). In addition, the spatial structure and resolution as well as the hidden seasonality (10 years climatologies are used) of the biogeochemical models might explain these discrepancies.

Minor comments

Add "sunlit" in the title". Also, as mentioned before, I would not use "functional convergence" as the main message of the paper

We replaced "global" by "sunlit" in the title. As described in more details previously, the overall shape of the clustering does indicate a functional convergence of distantly related lineages (now supported by further analyses), which we consider is an important observation that can only emerge from the analysis of a wide range of unicellular eukaryotic genomes. Our database provides an opportunity to apply such

method at a new scale. We consider our title summarizes the main insights of our study and modifying it would in our view minimize substantially its impact.

The number of metagenomes included in the analysis varied from 943 (page 14), to 939 (Table S1), and to 798 (in page 3). Please clarify.

The number “943” in page 14 was an error, now replaced by “939”. We thank the reviewer for finding this error.

As a clarification regarding “939” versus “798” metagenomes, we have used a total of 939 metagenomes in this study (as described in the Table S1), however only 798 metagenomes were used to generate the eukaryotic MAGs. Simply, the 0.2–3 μm size fraction was excluded for genome-resolved metagenomics because eukaryotic MAGs were already characterized from those metagenomes (see <https://www.nature.com/articles/s41564-018-0176-9>). This is explained in the main text, page 3:

“We performed the first comprehensive genome-resolved metagenomic survey of microbial eukaryotes from polar, temperate, and tropical sunlit oceans using 798 metagenomes (265 of which were released through the present study) derived from the Tara Oceans expeditions. They correspond to the surface and deep chlorophyll maximum layer of 143 stations from the Pacific, Atlantic, Indian, Arctic, and Southern Oceans, as well as the Mediterranean and Red Seas, encompassing eight eukaryote-enriched plankton size fractions ranging from 0.8 μm to 2 mm (Figure 1, Table S1). We used the 280 billion reads as inputs for 11 metagenomic co-assemblies (6–38 billion reads per co-assembly) using geographically bounded samples (Figure 1, Table S2), as previously done for the Tara Oceans 0.2–3 μm size fraction enriched in bacterial cells (ref 27).”

All size fractions were eventually used to determine the biogeography of the MAGs and SAGs.

Line 140. I could not find these three unclassified MAGs in Table S3

Indeed their identification was not clear enough. We have modified the Table S3, and taxonomy for those 3 MAGs is now labelled as “Putative_new_group”. We thank the reviewer for the careful examination of our tables.

Lines 250–251. I do not understand the logics of the “sister clades” to chrysophytes, Phaeocystis, or Pycnococcus. These MAGs may simply represent species from the group without representative genomes.

Many MAGs were connected to known lineages in the phylogenetic analyses we have performed. For example, we linked 20 MAGs to *Micromonas* because they were clearly connected to cultured species in our phylogenies. However, some MAGs and SAGs created distinct phylogenetic clades branching next to a known lineage. We observed this for Chrysophytes, *Phaeocystis*, and *Pycnococcus*. Phylogenetic signal indicates they fall outside the known family or genus, and awaiting a proper naming we have simply named them sister clades to the lineage they branch next to. This is a rather common approach.

Line 252. There are multiple MAST clades, and some of them are close to MAST-4.

Why all of them are interpreted as MAST-4?

We thank the reviewer for identifying this important issue. We have better clarified the taxonomic affiliation of MAST MAGs and SAGs (those contain the 18S rRNA gene marker), and slightly modified the main figure 2 accordingly. We also created the supplemental figure 7 to describe in more details the diversity of MAST lineages in two distinct c clusters (one includes MAST-3 while the other one includes MAST-4 and MAST-7).

This new figure and a reference to the article “Exploring the uncultured microeukaryote majority in the oceans: reevaluation of ribogroups within stramenopiles” have been incorporated into to the main text.

Lines 261-265. Truly dinoflagellates is an important group missing from the MAG collection, but it is not the only one. Other relatively abundant groups not found here are MALVs, acantharea, radiolaria, other MASTs, Telonema, Picozoa, diplomonids. You could analyze the Tara metabarcoding data (including multiple size fractions) to better identify what is missing in your list.

While we do have MALV MAGs and SAGs, other lineages the reviewer listed are indeed missing in our database. We have updated the main text to address this comment (**in bold**):

*“One of the most conspicuous lineages lacking any MAGs and SAGs was the Dinoflagellata, a prominent and extremely diverse phylum in small and large eukaryotic size fractions of Tara Oceans⁸. These organisms harbor very large and complex genomes⁵¹ that likely require much deeper sequencing efforts to be recovered by genome-resolved metagenomics. **Besides, many other important lineages are also missing in MAGs and SAGs (e.g., within Radiolaria and Excavata), possibly due to a lack of abundant populations despite their diversity.**”*

As a clarification, we do not claim that our database (<1,000 units) covers most plankton eukaryotic lineages. We would have been motivated to effectively link our MAGs and SAGs to 18S rRNA units identified from the Tara metabarcoding data.

Unfortunately, the SAGs and MAGs are depleted of the long, multi-copy, and evolutionary stable 18S rRNA genes, as already acknowledged in the text:

“Absent from the MAGs and SAGs are DNA molecules physically associated with the focal eukaryotic populations, but that did not correlate with their nuclear genomes across metagenomes. They include chloroplasts, mitochondria, and viruses generally present in multi-copy. Finally, some highly conserved multi-copy genes such as the 18S rRNA gene were also missing due to technical issues associated with assembly and binning, following the fate of 16S rRNA genes in bacterial MAGs (ref 27).”

Lines 330-357. It is nice to identify gene functions for clades C and D. Could this be done for clades A and B as well?

Table S5 displays the 3,268 differentially occurring functions between the four groups. As described in the main text, a wide range of functions is associated with

each group. Instead of describing a list of unrelated functions associated with each group, we searched for functions that appear to be connected and enriched or depleted in the same group(s). We found good examples for the groups C and D. We could not find good examples in groups A and B, and we have decided not to include randomly selected examples in order to avoid extending an already long manuscript.

Lines 470-473. I disagree here. Part of the functional grouping is driven by the "evolutionary history" (as commented above), and part is also driven by trophic modes (i.e., clade C seem obligate photoautotrophs).

In our view, this important phenomenon is already sufficiently explained in our manuscript. First, we wrote that part of the clustering is driven by "evolutionary history", but only when looking at the **fine-grained functional clusters**:

*"Fine-grained functional clusters exhibited a highly coherent taxonomy within the unicellular eukaryotes. For instance, MAGs affiliated to the coccolithophore *Emiliana* (completion ranging from 7.8% to 32.2%), Dictyochophaceae family (completion ranging from 8.6% to 76.9%) and the sister clade to *Phaeocystis* (completion ranging from 18.4% to 60.4%) formed distinct clusters. The sister clade to Cryptophyta (completion ranging from 1.6% to 75.7%) was also confined to a single cluster that could be explained partly by a considerable radiation of genes related to dioxygenase activity (up to 644 genes). Most strikingly, the Archaeplastida MAGs not only clustered with respect to their genuslevel taxonomy, but the organization of these clusters was highly coherent with their evolutionary relationships (see Figure 2), confirming not only the novelty of the putative sister clade to *Pycnococcus*, but also the sensitivity of our framework to draw the functional landscape of unicellular marine eukaryotes."*

In the following paragraph, we then cover the trophic mode (bold section was added to demonstrate that genomic incompleteness was not impacting the emergence of the four functional groups):

*"Four major functional groups of unicellular eukaryotes emerged from the hierarchical clustering (Figure 3), which was **perfectly recapitulated when** incorporating the standard culture genomes matching to a MAG (Figure S9), and when **clustering only the MAGs and SAGs >25% complete (Figure S10)**. Importantly, the taxonomic coherence observed in fine-grained clusters vanished when moving towards the root of these functional groups. Group A was an exception since it only covered the *Haptista* (including the highly cosmopolitan sister clade to *Phaeocystis*). Group B, on the other hand, encompassed a highly diverse and polyphyletic group of distantly related heterotrophic (e.g., MAST-4 and MALV) and mixotrophic (e.g., Myzozoa and Cryptophyta) lineages of various genomic size, suggesting that broad genomic functional trends may not only be explained by the trophic mode of plankton. Group C was mostly photosynthetic and covered the diatoms (Stramenopiles of various genomic size) and Archaeplastida (small genomes) as sister clusters. This finding likely reflects that diatoms are the only group with an obligatory photoautotrophic*

lifestyle within the Stramenopiles, like the Archaeplastida. Finally, Group D encompassed three distantly related lineages of heterotrophs (those systematically lacked gene markers for photosynthesis) exhibiting rather large genomes: Oomycota, Acanthoecida choanoflagellates, and the Cryptophyta's sister clade."

As emphasized in the figures 2 and 3 ("phytoplankton gene markers" layer), the functional clustering cannot be explained by evolution and trophic mode. This is why we elected to focus on this interesting insight into the functioning of planktonic unicellular eukaryotic lineages.

We thank the reviewer for significantly improving the strength of our manuscript.

Referees' report, second round of review

Reviewer #1 (Comments to authors)

The authors have thoroughly responded to our concerns and have strengthened the manuscript.

Reviewer #2 (Comments to authors)

This is my second view of a paper that I initially liked, but I raised few concerns, doubts and suggestions. I am satisfied to see that most of my concerns have been extensively addressed, as well as some of the suggestions for improvement have been accommodated in the revised version. This manuscript represents an impressive amount of data and analysis, and the resubmitted version presents a more convincing and clearer story. I only have a few comments to be taken in consideration.

Regarding functional convergence, lines 39-41 say: Neither trophic modes of plankton nor its vertical evolutionary history could explain the functional repertoire convergence of major eukaryotic lineages

This is part of what I mentioned in the previous version: I guess the functional grouping can be explained partially by the 1) trophic modes, 2) vertical evolutionary history, and 3) independent functional convergence. This is different of what is said in the text, that states that reasons 1) and 2) do not play a role. For me a fairest way to explain this would be something like "Trophic modes and vertical evolutionary history do not completely explain the functional groups, so an extent of functional convergence is at play"

Lines 119-121. I appreciate that you removed the term SMAGs at the new version. Still for all this long section (up to line 274), I do not see why mixing the MAGs and SAGs analysis. Both genomic resources derive from very, very different approaches. As said in my first round of review, I would focus in this part on the MAGs. And at the end, you could explain what SAGs were adding to the previous picture

Line 130-132. I don't understand this logics for chloroplast (and probably for mitochondria), as generally there is a fixed number of organelles within a cell of each species. For instance, prymnesiophytes tend to have 2 chloroplasts, mamiellophytes 1 chloroplast.

Line 140. Perhaps add one decimal to the "0%"

Line 261-262. I do not see Picozoa in this study, even though these are important marine protists. Perhaps these are within the unclassified signals (perhaps the new Archaeplastida). Recently available picozoan SAGs can be used to address this question (Schön, et al. 2021. Single cell genomics reveals plastid-lacking Picozoa are close relatives of red algae. Nat Commun 12, 6651).

Authors' response to the second round of review

Author responses to reviewer #1

The authors have thoroughly responded to our concerns and have strengthened the manuscript.

We thank the reviewer for contributing to the strength of to our study.

Author responses to reviewer #2

This is my second view of a paper that I initially liked, but I raised few concerns, doubts and suggestions. I am satisfied to see that most of my concerns have been extensively addressed, as well as some of the suggestions for improvement have been accommodated in the revised version. This manuscript represents an impressive amount of data and analysis, and the resubmitted version presents a more convincing and clearer story. I only have a few comments to be taken in consideration.

Regarding functional convergence, lines 39-41 say: Neither trophic modes of plankton nor its vertical evolutionary history could explain the functional repertoire convergence of major eukaryotic lineages

This is part of what I mentioned in the previous version: I guess the functional grouping can be explained partially by the 1) trophic modes, 2) vertical evolutionary history, and 3) independent functional convergence. This is different of what is said in the text, that states that reasons 1) and 2) do not play a role. For me a fairest way to explain this would be something like "Trophic modes and vertical evolutionary history do not completely explain the functional groups, so an extent of functional convergence is at play"

We thank the reviewer for sharing their thoughts on this important topic, at the center of our study. We agree that all three mechanisms outlined in this comment are likely at play, albeit at different levels of importance. As described in detail in the manuscript, the clustering of genomes based on the occurrence of functions (functional profiles, Figure 3) connects distantly related lineages for 3 out of the 4 main clusters. Thus, vertical evolutionary history appears not to be the main factor. Then, since clusters A, B and C all contain photosynthetic organisms, trophic mode appears also not to be the main factor (photosynthetic organisms are split into different groups). This is why data we generated points to functional divergences of distantly related lineages (those cover various trophic modes) representing a key mechanism for the occurrence of functions across the tree of marine microbial

eukaryotic life. This is stated in the abstract and elsewhere in the manuscript, and is in our humble view an important insight from our genome-scale functional analysis of eukaryotic plankton. As a result, we consider that our sentence in the abstract is coherent with the data, analyses and interpretations present in our manuscript. Nevertheless, we added “**completely**” to address, at least to some extent, the reviewer’s comment:

*“Neither trophic modes of plankton nor its vertical evolutionary history could **completely** explain the functional repertoire convergence of major eukaryotic lineages that coexisted within oceanic currents for millions of years.”*

Again, we do not claim that trophic modes and the vertical evolutionary history have no impact. This is clarified in different parts of the manuscript. We are confident the reviewer understands our reasoning not to minimize the effect of functional divergence in the abstract, given the patterns we observed in figure 3 and other supplemental figures.

Lines 119-121. I appreciate that you removed the term SMAGs at the new version. Still for all this long section (up to line 274), I do not see why mixing the MAGs and SAGs analysis. Both genomic resources derive from very, very different approaches. As said in my first round of review, I would focus in this part on the MAGs. And at the end, you could explain what SAGs were adding to the previous picture

Metagenome-assembled genomes, culture genomes and single-cell genomes indeed have very different methodologies but all provide an important genomic context to explore the functioning, biogeography, and evolution of eukaryotic plankton. In our study, we used MAGs and SAGs generated within the scope of *Tara* Oceans to fill gaps in our culture portfolio. Critically, excluding SAGs in our phylogenetic analysis would have prevented determining their taxonomic relevance in the context of cultures and MAGs. Indeed, we learned from this analysis that SAGs provided access to lineages not covered by MAGs. The two resources were therefore complementary, which is likely a point of interest to others in our field. We hope the reviewer understands our reasoning not to exclude SAGs in the first sections of our study, which cover especially the phylogenetic analyses.

Line 130-132. I don't understand this logics for chloroplast (and probably for mitochondria), as generally there is a fixed number of organelles within a cell of each species. For instance, prymnesiophytes tend to have 2 chloroplasts, mamiellophytes 1 chloroplast.

We thank the reviewer for pointing out that chloroplasts may have a fixed copy number for each eukaryotic population. In the present study, we found that eukaryotic MAGs were depleted of chloroplast and mitochondria. In a parallel ongoing study, we found that chloroplast MAGs were identified in different metabins (the CONCOCT clusters characterized by means of constrained automatic binning) as compared to the eukaryotic MAGs (data not shown). Our work on chloroplast MAGs is not yet completed. However, we have learned already that

those chloroplasts MAGs are distinct from the eukaryotic MAGs when considering sequence composition and differential coverage. This explains why they are not present in the eukaryotic genomic database presented in the present study. We have modified the text for clarity and to address this point (see **bold section**):

*“Absent from the MAGs and SAGs are DNA molecules physically associated with the focal eukaryotic populations, but that did not **necessarily** correlate with their nuclear genomes across metagenomes or had **distinct sequence composition.**”*

Line 261-262. I do not see Picozoa in this study, even though these are important marine protists. Perhaps these are within the unclassified signals (perhaps the new Archaeplastida). Recently available picozoan SAGs can be used to address this question (Schön, et al. 2021. Single cell genomics reveals plastid-lacking Picozoa are close relatives of red algae. Nat Commun 12, 6651).

We are very grateful to the reviewer for pointing to the recent study. We integrated the Picozoa SAGs into our phylogenetic framework of RNA polymerase genes, which revealed that what was initially named “sister clade of Cryptophyta” in fact corresponds to the phylum Picozoa. We have modified the text, figures and tables accordingly. Again, we thank the reviewer very much for improving our taxonomic inference of eukaryotic MAGs, and for the previous contributions made to our study.